# Newly Isolated Virulent Salmophages for Biocontrol of Multidrug-Resistant *Salmonella* in Ready-to-Eat Plant-Based Food

**DOI:** 10.3390/ijms241210134

**Published:** 2023-06-14

**Authors:** Michał Wójcicki, Olga Świder, Paulina Średnicka, Dziyana Shymialevich, Tomasz Ilczuk, Łukasz Koperski, Hanna Cieślak, Barbara Sokołowska, Edyta Juszczuk-Kubiak

**Affiliations:** 1Laboratory of Biotechnology and Molecular Engineering, Department of Microbiology, Prof. Waclaw Dabrowski Institute of Agricultural and Food Biotechnology—State Research Institute, Rakowiecka 36 Str., 02-532 Warsaw, Poland; paulina.srednicka@ibprs.pl (P.Ś.); edyta.juszczuk-kubiak@ibprs.pl (E.J.-K.); 2Department of Food Safety and Chemical Analysis, Prof. Waclaw Dabrowski Institute of Agricultural and Food Biotechnology—State Research Institute, Rakowiecka 36 Str., 02-532 Warsaw, Poland; olga.swider@ibprs.pl; 3Culture Collection of Industrial Microorganisms—Microbiological Resources Center, Department of Microbiology, Prof. Waclaw Dabrowski Institute of Agricultural and Food Biotechnology—State Research Institute, Rakowiecka 36 Str., 02-532 Warsaw, Poland; diana.szymielewicz@ibprs.pl (D.S.); hanna.cieslak@ibprs.pl (H.C.); 4Department of Pathology, Medical University of Warsaw, Pawińskiego 7 Str., 02-106 Warsaw, Poland; tomasz.ilczuk@wum.edu.pl (T.I.); lukasz.koperski@wum.edu.pl (Ł.K.); 5Department of Microbiology, Prof. Waclaw Dabrowski Institute of Agricultural and Food Biotechnology—State Research Institute, Rakowiecka 36 Str., 02-532 Warsaw, Poland

**Keywords:** virulent bacteriophages, *Salmonella enterica*, salmonellosis, multidrug resistance, genomic analysis, functional annotation, food preservation, minimally processed food

## Abstract

Due to irrational antibiotic stewardship, an increase in the incidence of multidrug resistance of bacteria has been observed recently. Therefore, the search for new therapeutic methods for pathogen infection treatment seems to be necessary. One of the possibilities is the utilization of bacteriophages (phages)—the natural enemies of bacteria. Thus, this study is aimed at the genomic and functional characterization of two newly isolated phages targeting MDR *Salmonella enterica* strains and their efficacy in salmonellosis biocontrol in raw carrot–apple juice. The *Salmonella* phage vB_Sen-IAFB3829 (*Salmonella* phage strain KKP 3829) and *Salmonella* phage vB_Sen-IAFB3830 (*Salmonella* phage strain KKP 3830) were isolated against *S*. I (6,8:l,-:1,7) strain KKP 1762 and *S*. Typhimurium strain KKP 3080 host strains, respectively. Based on the transmission electron microscopy (TEM) and whole-genome sequencing (WGS) analyses, the viruses were identified as members of tailed bacteriophages from the *Caudoviricetes* class. Genome sequencing revealed that these phages have linear double-stranded DNA and sizes of 58,992 bp (vB_Sen-IAFB3829) and 50,514 bp (vB_Sen-IAFB3830). Phages retained their activity in a wide range of temperatures (from −20 °C to 60 °C) and active acidity values (pH from 3 to 11). The exposure of phages to UV radiation significantly decreased their activity in proportion to the exposure time. The application of phages to the food matrices significantly reduced the level of *Salmonella* contamination compared to the control. Genome analysis showed that both phages do not encode virulence or toxin genes and can be classified as virulent bacteriophages. Virulent characteristics and no possible pathogen factors make examined phages feasible to be potential candidates for food biocontrol.

## 1. Introduction

The human microbiome plays a significant role in the state of human health, and dysbiosis—the imbalance between beneficial and pathogenic microbes—affects the development of diseases in various organs, including the intestine [1,2,3]. One of the main bacterial pathogens transmitted through food and water is *Salmonella*—the etiological agent of diseases such as typhoid (caused by *S*. Typhi), paratyphoid (*S*. Paratyphi A, B, or C), or salmonellosis (various other *S. enterica* serovars) [4,5,6,7]. Due to irrational antibiotic stewardship and the lack of a proper antibiotic policy, an increase in the incidence of the multidrug resistance (MDR) of bacteria has been observed recently [7,8,9]. One of the vectors of antibiotic resistance is the food chain through which bacteria can acquire antibiotic resistance genes (ARGs) [10,11,12]. In Poland, *Salmonella* infections are still the most common cause of food poisoning. In the latest report by the Chief Sanitary Inspectorate for 2021, 8269 cases caused by these bacteria were confirmed, of which 7975 were (96.4%) related to food poisoning. The most common sources of collective outbreaks of food poisoning and foodborne infections were contaminated milk and other dairy products, ready meals, eggs and products containing them, pork and poultry meat, seafood, and secondarily contaminated food of plant origin [13]. Our research on antibiotic resistance profiles of *Salmonella* isolated from various links of the food chain in Poland showed that 50.9% of the tested strains were MDR [14].

Considering the above, the search for new therapeutic methods for foodborne pathogen infection treatment is necessary. One of the possibilities is the utilization of bacteriophages—natural enemies of bacteria [15,16,17]. Despite the lack of approval for the use of preparations based on lytic bacteriophages in the European Union, many research centers have conducted research on the development of effective phage biopreparations [18,19,20]. This biological food preservation method has been approved for use in several non-EU countries, such as the USA, Switzerland, and Canada [18,21]. Several biopreparations targeted at the elimination of *Salmonella*, i.e., SalmoFresh™ (Intralytix, Inc., Baltimore, MD, USA), PhageGuard S™ (Micreos Food Safety, Wageningen, The Netherlands), and SalmoPro^®^ (Phagelux, Shanghai, China) are commercially available. All of them are approved by the Food and Drug Administration (FDA) [21,22,23]. In addition, the Polish company, Proteon Pharmaceuticals S.A. (Lodz, Poland), also develops and commercializes phage biopreparations for use as feed additives for animal husbandry [24]. The preparation under the trade name Bafasal^®^, at the request of the European Commission, received a positive opinion from the Panel on Additives and Products or Substances Used in Animal Feed (FEEDAP). Bafasal^®^ is a preparation containing four strictly lytic bacteriophages targeting *Salmonella* sp., intended for use as a zootechnical additive to drinking water and liquid complementary feed for all species of birds [25].

It is worth noting that phage preparations are highly specific agents, often active only against a specific species or bacterial strain of the host [26,27]. In view of the above, if such agents are approved for use in the future, it will be important to develop a biopreparation method effective against autochthonous strains (serovars) of *Salmonella*.

Thus, this study is aimed at the genomic and functional characterization of two newly isolated bacteriophages targeting MDR *Salmonella enterica* strains and their efficacy in salmonellosis biocontrol in ready-to-eat plant-based food.

## 2. Results and Discussion

### 2.1. Bacterial Host Strains

All bacterial strains used in our research came from the Culture Collection of Industrial Microorganisms—Microbiological Resources Center of the Department of Microbiology at the Prof. Waclaw Dabrowski Institute of Agricultural and Food Biotechnology—State Research Institute (IAFB; Warsaw, Poland). Two multidrug-resistant (MDR) strains of *Salmonella* were used for the isolation of bacteriophages, i.e., *S*. I (6,8:l,-:1,7) strain KKP 1762, and *S*. Typhimurium strain KKP 3080. Data on the phenotypic profiles of antibiotic resistance and the presence of antibiotic resistance genes were presented in the previous article [14]. *Salmonella* strains were isolated from food products in 2010 and 2019 (Table 1). 

*Salmonella enterica* subsp. *enterica* serovar Typhimurium strain KKP 3080 exhibited a resistance profile to eleven antibiotics, belonging to four different classes of antibiotics (penicillins, cephalosporins, fluoroquinolones, and aminoglycosides) [14]. The second strain, *Salmonella enterica* subsp. *enterica* serovar 6,8:l,-:1,7 strain KKP 1762, had a similar antibiotic susceptibility profile, but did not show phenotypic resistance to cefepime, cefotaxime, and ofloxacin.

The genomes of the host bacterial strains were sequenced and the full assembled genomes were deposited in the GenBank database under accession numbers CP121297 (*S*. I (6,8:l,-:1,7) strain KKP 1762) and CP121298 (*S*. Typhimurium strain KKP 3080). Table 2 and Figure 1, Figure 2 and Figure 3 show maps of the bacterial host genomes’ organization.

The *S*. I (6,8:l,-:1,7) strain KKP 1762 genome is 4,578,573 bp length with a G+C content of 52.2% (Table 2). Forty antibiotic resistance genes (ARGs) have been identified in the genome of this strain. Seventy-four putative horizontal gene transfer (HGT) events were predicted. In addition, 315 regions related to mobile genetic elements (MGEs) were identified, of which 103 related to phage, 40 related to transfer, 34 related to integration/excision, 113 related to replication/recombination/repair, and 25 related to stability/transfer/defense. Moreover, two CRISPR arrays have been predicted; however, Cas proteins which are associated with them have not been found. In *S*. I (6,8:l,-:1,7) strain KKP 1762 genome, four regions (136,125 bp, 11,738 bp, 70,211 bp, and 160,306 bp, respectively) and 156 prophage genes were detected, including 3 related to integration (Figure 1). A prophage is an inherited form of a temperate bacteriophage, functioning as a dormant form of the viral genome, which is mostly integrated into the bacterial chromosome and replicates vertically with the genome of the bacterial host [28,29]. In bacterial cells, after infection with lysogenic phage, phenotypic changes (lysogenic conversion) are observed [28,30]. Lysogenic conversion may result in a resistance to related phages and in some cases, the acquisition of cargo genes affects population dynamics, including genes involved in virulence, metabolism, or antibiotic resistance genes [30]. Until the lytic cycle resumes, no offspring are produced. Prophages can be defective or functional. Defective prophages have lost part of their genome and can no longer produce daughter particles but may still be able to excise from the host chromosome. In turn, a prophage is considered functional if it can resume the lytic cycle and re-infect host cells [28].

**Table 2 ijms-24-10134-t002:** The *Salmonella* genomes’ sequence details obtained with Prokka 1.14.6 v1.1.0 software [31].

*Salmonella* Strain Number	Genome Length	G+C Content	CDS	Gene/mRNA	rRNA	Repeat Region	tRNA	tmRNA
KKP 1762	4,578,573 bp	52.2%	4265	4385	22	2	87	1
KKP 3080	4,777,899 bp	52.2%	4439	4554	22	2	84	1

**Figure 1 ijms-24-10134-f001:**
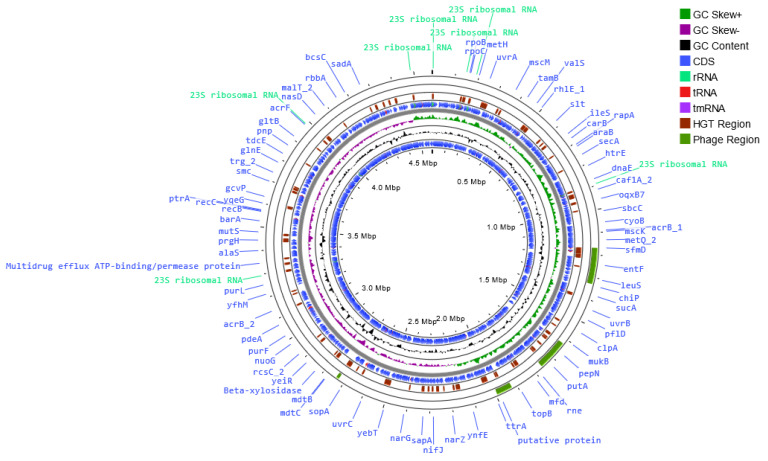
Map of the genome organization of *Salmonella enterica* subsp. *enterica* serovar 6,8:l,-:1,7 strain KKP 1762 generated using the Proksee software [32].

**Figure 2 ijms-24-10134-f002:**
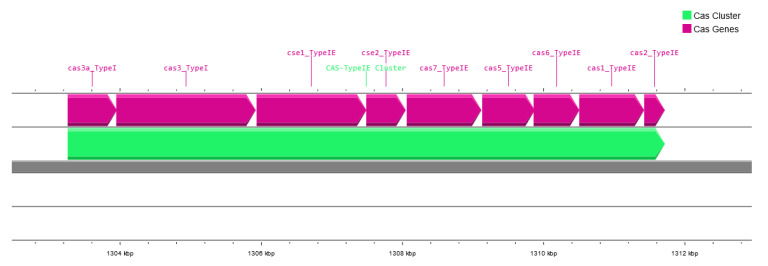
The CRISPR features in *Salmonella enterica* subsp. *enterica* serovar Typhimurium strain KKP 3080 generated using the Proksee software [32].

**Figure 3 ijms-24-10134-f003:**
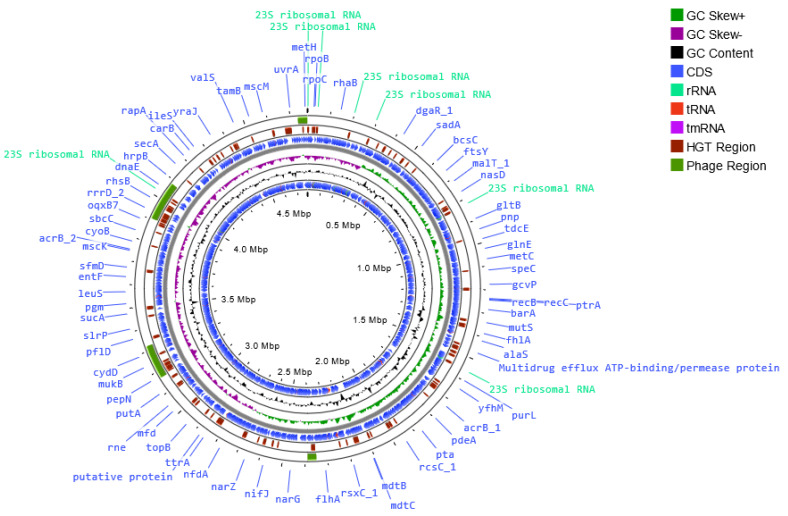
Map of the genome organization of *Salmonella enterica* subsp. *enterica* serovar Typhimurium strain KKP 3080 generated using the Proksee software [32].

Prophage genes contribute extensively to host physiology. Many studies have thus far been involved in the identification and characterization of the virulence factors of pathogenic bacteria encoded by phages [33,34,35]. Horizontally acquired prophage genes are important factors in genomic evolution and provide discrete adaptive physiological contributions, such as enhancing fitness under specific environmental conditions or providing non-obvious metabolic or signaling functions [28,35]. 

The *S*. Typhimurium strain KKP 3080 genome is 4,777,899 bp in length, with a G+C content of 52.2% (Table 2). Forty-two ARGs have been identified, and eighty-four putative HGT events were predicted in its genome. Moreover, 341 regions related to MGEs were identified, of which 108 related to phage, 43 related to transfer, 40 related to integration/excision, 109 related to replication/recombination/repair, and 41 related to stability/transfer/defense. In addition, four CRISPR arrays (101 bp, 1188 bp, 1018 bp, and 110 bp, respectively), nine Cas genes, and one cluster (8461 bp) have been predicted (Figure 2). 

The clustered regularly interspaced short palindromic repeats (CRISPR)–Cas system confers adaptive resistance to bacteria against invasion by MGEs, including viruses, plasmids, and transposons [36,37]. The genus *Salmonella* is known to carry a class 1 type I–E system, closely related to the CRISPR–Cas system in *Escherichia coli* [38]. The systems have been reported to carry either one or two CRISPR loci and *cas*-gene clusters of *cas3, cse1-cse2-cas7-cas5-cas6e-cas1-cas2* genes [38,39]. This system captures protospacers from invading MGEs and incorporates them into the CRISPR array using Cas proteins. CRISPR–Cas is also involved in alternative functions such as the management of bacterial virulence and physiology. In some species of bacteria, including *Salmonella*, selective protospacers were found in the bacterial genome, which confirms the role of the CRISPR–Cas system in the regulation of endogenous genes [36]. 

In the *S*. Typhimurium strain KKP 3080 genome, four regions (41,418 bp, 176,857 bp, 151,444 bp, and 39,063 bp, respectively), and 154 prophage genes were detected, including 2 related to integration (Figure 3).

### 2.2. Bacteriophages and Their Host Range

Target phages were screened against two selected multidrug-resistant *Salmonella* strains in raw municipal wastewater. The *Salmonella* phage strain KKP 3829 and *Salmonella* phage strain KKP 3830 were isolated against *S*. I (6,8:l,-:1,7) strain KKP 1762 and *S*. Typhimurium strain KKP 3080 host strains, respectively. The purified and amplified phage lysates (phage titer ~10^7^ PFU mL^−1^) were used to define the bacterial host range by spot test (Table 3).

Most of the observed phage lysis zones were turbid plaques. *Salmonella* phage strain KKP 3829 showed a wide range of bacterial hosts. It infected 92.6% (50/54) of *Salmonella* strains. In addition, it showed activity against three strains of bacteria from the *Enterobacter cloacae* species and three from the *Escherichia coli* species. *Salmonella* phage strain KKP 3830 showed a much smaller range of activity (44.4% (24/54) against *Salmonella* strains). At the same time, it did not show activity against other non-pathogenic *Enterobacterales*. None of the isolated phages showed activity against four strains of the *Salmonella* genus, i.e., *S*. Derby strain KKP 1006, rough *S. enterica* strain KKP 1113, *S*. Mbandaka strain KKP 1169, and *S. Senftenberg* strain KKP 1597. Moreover, the phages did not show activity against the tested bacteria of the *Citrobacter*, *Pantoea*, *Raoultella*, and *Serratia* genera. Pathogenic bacteria other than *Salmonella*, both Gram-negative (*Pseudomonas aeruginosa*) and Gram-positive (*Listeria monocytogenes*, *Staphylococcus aureus*), were non-susceptible to the tested phages.

The assessment of the host range for phages is an important step in terms of the characterization of bacteriophages that are potential candidates to be used in the developed biopreparation [40,41]. Phage cocktails intended for phage therapy are typically created by combining several known or newly isolated phages to create a therapeutic mixture with the broadest possible host specificity [40,41,42,43]. Mixtures of different phages with a wide host range can be an alternative/equivalent to broad-spectrum antibiotics [44,45]. Two newly isolated bacteriophages overlap our host range entirely. Despite this, the inclusion of both bacteriophages in the phage biopreparation may provide a greater biodiversity and improve its effectiveness, e.g., in the case of the host acquiring resistance to one of the phages.

### 2.3. One-Step Growth of Phages

To determine the growth characteristics of phages, one-step growth curves were analyzed (Figure 4). For both phages, the latent period was 20 min. The rise period (burst time) for the tested phages was at a similar level, from 55 min (*Salmonella* phage strain KKP 3830) to 65 min (*Salmonella* phage strain KKP 3829). The phage burst size was at a similar level, from 11 ± 1 PFU cell^−1^ (*Salmonella* phage strain KKP 3830) to 22 ± 0 PFU cell^−1^ (*Salmonella* phage strain KKP 3829) (Table 4). In other studies, concerning salmophages, similar latent periods but different burst sizes were reported. In the study by Islam et al. [46], the latent period for salmophage LPST153 was 10 min, and the burst size was 113 ± 8 PFU cell^−1^. In turn, in the study by Abdelsattar et al. [47], the latent period for the salmophage ZCSE9 was 10 min, but the burst size was about 20 PFU cell^−1^.

Both latent period and burst size of the phages are key factors when considering whether a phage may be selected for biological control experiments [48]. It has been proven that a large burst size and short latent period are positively correlated with effective bacterial inactivation [49].

### 2.4. Phage Adsorption to Host Bacterial Cells

Adsorption experiments revealed that each of the tested phages adsorbed differently to its bacterial host cells (Figure 5). Using MOI = 0.1 at 37 °C after 5 min of incubation, adsorption levels averaged 30.0%, and 21.1% for *Salmonella* phage strain KKP 3829, and *Salmonella* phage strain KKP 3830, respectively. At the end of the experiment (20 min), both phages significantly increased the amount of their particles adsorbed to the bacterial host cells.

The determined adsorption rate constants *k* ranged from 1.15 × 10^9^ mL min^−1^ (*Salmonella* phage strain KKP 3830) to 2.69 × 10^9^ mL min^−1^ (*Salmonella* phage strain KKP 3829). The adsorption rate constant *k* correlated with the experimentally determined adsorption coefficients—phages with the highest *k* values (i.e., *Salmonella* phage strain KKP 3829) achieved a high adsorption coefficient (>70%) at the end of the experiments (20 min) (Table 5).

In the study by Abdelsattar et al. [47], it was shown that the time necessary for adsorption of the maximum number of phages was almost 20 min for salmophage ZCSE9. Salmophage ZCSE9, in contrast to our two newly isolated phages, adsorbed very rapidly—within two minutes, it adsorbed 70% of viruses.

### 2.5. Growth Kinetics of Bacterial Hosts after Phage Infection

The lytic activity of phages was determined using a Bioscreen C Pro automated growth analyzer. To this end, growth curves were plotted for each bacterial host strain (optical density in the function of bacteria cell number; data unpublished). The determination of an equation of a straight line allowed the determination of the optical density values for bacterial strain cultures, which in turn enabled the adjustment of the appropriate values of the multiplicity of infection coefficient for each phage (MOI = 1000; MOI = 100; MOI = 10; MOI = 1.0; MOI = 0.1; MOI = 0.01; MOI = 0.001; and MOI = 0.0001, respectively). The results showed that phages, depending on the MOI used, inhibited the growth of bacterial hosts to a varying extent (Figure 6).

Optical density measurement made with the Bioscreen C Pro automated growth analyzer allowed us to establish the onset and duration of the logarithmic stage of growth of the bacterial strains, which were deliberately infected with specific phages at different MOIs compared to the control culture. For both *Salmonella* strains, infection with the targeted phage at each MOIs caused a significant decrease (*p* ≤ 0.0001) in the optical density of the culture compared to the control culture. For *S*. I (6.8:1,-:1.7) strain KKP 1762 treated with the target phage, i.e., *Salmonella* strain KKP 3829, using the four highest MOIs (i.e., MOI = 1000; MOI = 100; MOI = 10; and MOI = 1.0), the growth of bacterial hosts was completely inhibited. The addition of phages to the cultures of *S*. Typhimurium strain KKP 3080 resulted in a delayed start of the logarithmic growth phase of the hosts compared to the control culture. Regardless of the MOIs used, host growth curves were similar in appearance, and even the highest MOIs used did not completely inhibit host growth.

The changes in the optical density of the tested bacterial strains after incubation with specific phages are presented in Table 6. The lower coefficients of the specific growth rate (μ) determined for the phage-infected cultures indicate a significant suppression of cell division during the logarithmic stage of growth of the *Salmonella* strains.

In the study conducted by Islam et al. [46], salmophage LPST153 could consistently inhibit the growth of *S*. Typhimurium ATCC 13311 with fewer counts at MOI ratios of 0.1, 1, 10, and 100 over 12 h. In turn, the bactericidal activity of salmophage ZCSE9 was improved by increasing the MOI from 0.0001 to 10 [47]. Previous analyses revealed that a phage is more effective at reducing the host when used with high MOIs [50,51]. In the study conducted by Mahmoud et al. [52], the growth of *Salmonella* Kentucky infected with bacteriophages at MOI 1.0 was delayed by all tested phages compared to control cultures. Moreover, a complete inhibition of the host growth was observed after 24 h of incubation with target phages.

### 2.6. Determination of Morphological Features of Phages and Their Plaques

*Salmonella* phage strain KKP 3829 formed small transparent plaques below 1 mm in diameter, while *Salmonella* phage strain KKP 3830 formed transparent plaques ~2 mm in diameter. It is generally considered that smaller-sized phages diffuse more easily in the agar layer and thus form larger plaques [53,54]. In the case of our two newly isolated phages, we did not observe this relationship. None of the phages produced a clearly visible ‘halo’ zone (Figure 7, left side). It is considered that the phenotype of the ‘halo’ zone is an indication of the ability to depolymerize exopolysaccharide (by producing EPS depolymerases) and can be effective in dispersing biofilms [55]. Electronograms obtained with the use of TEM allowed us to visualize the virions morphology (Figure 7, right side). Both phages have a complex structure (tailed phages) containing a head and long non-contractile tails. *Salmonella* phage strain KKP 3829 has an elongated capsid, with a length of 30.8 nm, and is a small salmophage with a total length of approximately 80.3 nm (Table 7). On the other hand, *Salmonella* phage strain KKP 3830 has an isometric head, with a length of 70.4 nm. Salmophage strain KKP 3830 has a long tail, measuring 154.5 nm long by 12.3 nm wide (Figure 7 and Table 7).

### 2.7. Phage Genome Sequencing and Bioinformatics Analysis

The complete genomes of the *Salmonella* phage strain KKP 3829 and *Salmonella* phage strain KKP 3830 have been sequenced and deposited in the GenBank database under the accession numbers OQ674105 and OQ674106, respectively. Moreover, the newly isolated phages were deposited in the Culture Collection of Industrial Microorganisms—Microbiological Resources Center of the Department of Microbiology at the Prof. Waclaw Dabrowski Institute of Agricultural and Food Biotechnology—State Research Institute (IAFB; Warsaw, Poland), which is the potential partner of Microbial Resource Research Infrastructure—European Research Infrastructure Consortium (MIRRI–ERIC).

For the genomic characterization of the bacteriophages, we used the latest guidelines on the taxonomy of bacterial viruses published in January 2023 [56]. Similar to TEM, genome analysis confirmed that both salmophages belonged to the tailed phages from the *Caudoviricetes* class. Proteomic trees generated by the BIONJ program-based TBLASTX genomic sequence comparisons of other phage genomes deposited in the Virus–Host DB [57] are presented in Figure 8.

Genome sequencing of the *Salmonella* phage strain KKP 3829 and *Salmonella* phage strain KKP 3830 revealed that these phages have linear double-stranded DNA (dsDNA). A genome map of *Salmonella* phage strain KKP 3829 belonging to the *Casjensviridae* family is presented in Figure 9. *Salmonella* phage strain KKP 3829 has a genome length of 58,992 bp with a total G+C content of 56.5%. Out of the 168 predicted open reading frames (ORFs), 37 ORFs are associated with genes encoding proteins with known functions and 49 ORFs encode hypothetical proteins with unknown functions (Figure 9 and Appendix A). 

No potential tRNA is detected in the genome, suggesting that *Salmonella* phage strain KKP 3829 relies on the tRNA of its host for protein expression. The lack of tRNA genes can be associated with a highly compact phage genome that tends to lack any translational-associated genes to exclude nonessential information [59]. Based on the functional organization of proteins, the genes in the salmophage strain KKP 3829 genome could be divided into four groups: lysis, replication and metabolism-related, genome-packaging-related, and structural genes. Proteins associated with lysis include, among others, holin, lysozyme, and spanin. In the *Salmonella* phage strain KKP 3829 genome, three spanin coding regions (60, 84, and 112 amino acids length, respectively) were predicted. Spanins are phage lysis proteins required to disrupt the outer membrane. Phages use binary spanins or single-molecular spanins in the final step of lysing Gram-negative bacteria. Two-component spanins, such as Rz–Rz1 from phage lambda, are composed of an integral inner membrane protein, i-spanin, and an outer membrane lipoprotein, o-spanin, which form a periplasm-containing complex [60,61]. Endolysin (237 amino acids length) was predicted in the genome of *Salmonella* phage strain KKP 3829. One of the endolysins is lysozyme [62,63]. Double-stranded DNA phages use the holin–endolysin system to lyse bacterial host cells. Holin lyses the bacterial cell membrane. In this way, the host cell wall is exposed to lysozyme to degrade the bacterial peptidoglycan, followed by the release of progeny phages [64,65,66]. In addition, no lysogenic genes (i.e., encoded integrase, recombinase, repressors, or excisionase) are found in the *Salmonella* phage strain KKP 3829 genome, and no gene products show homology to a toxin or other virulence factors, suggesting that *Salmonella* phage strain KKP 3829 could be a virulent (lytic) and safe phage [62,67]. Among the genes related to replication and metabolism, in the *Salmonella* phage strain KKP 3829, the genome contains the following proteins: DNA helicase, DNA primase, DNA polymerase, HTH DNA binding protein, Gp2.5-like ssDNA binding protein and ssDNA annealing protein, endonuclease, and others. Terminase small and large subunits have been predicted in the *Salmonella* phage strain KKP 3829 genome as the proteins related to genome packaging. In the genome of this phage, genes encoding proteins related to the structure are also predicted: tail fiber protein, tail protein, major head protein, portal protein, and others (Appendix A).

BLASTn similarity searches performed for *Salmonella* phage strain KKP 3829 and related phages deposited in GenBank revealed a 96.13% nucleotide similarity with *Salmonella* phage FSL SP-124 (GenBank Acc. No. KC139515.1), 96.06% nucleotide similarity with *Salmonella* phage FSL SP-088 (GenBank Acc. No. NC_021780.1), 96.21% nucleotide similarity with *Salmonella* phage SPN19 (GenBank Acc. No. JN871591.1), 96.28% nucleotide similarity with *Salmonella* phage YSD1 strain YSD1_PHAGE (GenBank Acc. No. NC_048666.1), and 95.69% nucleotide similarity with *Salmonella* phage BSPM4 (GenBank Acc. No. NC_048655.1) (Figure 10, Appendix A).

In addition, *Salmonella* phage strain KKP 3829 was compared with other bacterial viruses reported in the PhageAI database. Phage similarity on a 2D scatter plot computationally predicted and rendered through PhageAI v.1.0.2 software is shown in the Appendix A (Appendix A). 

A genome map of *Salmonella* phage strain KKP 3830 belonging to the *Drexlerviridae* family is presented in Figure 11. *Salmonella* phage strain KKP 3830 has a genome length of 50,514 bp, with a total G+C content of 42.9%. The genome of *Salmonella* phage strain KKP 3830 had 102 predicted coding sequences (CDSs). Out of the 102 CDSs, 37 (36.3%) could be assigned putative functions, and 65 (63.7%) could not (Figure 11 and Appendix A). No potential tRNA is detected in the genome, suggesting that *Salmonella* phage strain KKP 3830, similar to *Salmonella* phage strain KKP 3829, relies on the tRNA of its host for protein expression.

Based on the functional organization of proteins, the genes in the salmophage strain KKP 3830 genome could be divided into four groups: lysis, replication and metabolism-related, genome-packaging-related, and structural genes. Both holin (71 amino acids length) and lysozyme (i.e., endolysin; 164 amino acids length) were predicted in the phage genome. In the *Salmonella* phage strain KKP 3830 genome, one spanin coding region (129 amino acids length) was predicted. In addition, no genes encoded integrase, recombinase, repressors, and excisionase, which are the main markers of lysogenic (temperate) viruses, are found in the *Salmonella* phage strain KKP 3830 genome. Moreover, no gene products show homology to a toxin or other virulence factors, suggesting that *Salmonella* phage strain KKP 3830 could be a strictly virulent phage. Among the genes related to replication and metabolism, in the *Salmonella* phage strain KKP 3830 genome, the following proteins were present: DNA helicase, DNA primase, DNA methyltransferase, single-strand DNA binding protein, Erf-like ssDNA annealing protein, and others. Terminase small and large subunits have been predicted in the *Salmonella* phage strain KKP 3830 genome as the proteins related to genome packaging. In the genome of this phage, genes encoding proteins related to the structure are also predicted: tail fiber protein, tail protein, major head protein, portal protein, and others (Appendix A).

BLASTn similarity searches performed for *Salmonella* phage strain KKP 3830 and related phages deposited in GenBank revealed a 99.74% nucleotide similarity with *Salmonella* phage YSP2 (GenBank Acc. No. NC_047898.1), 99.77% nucleotide similarity with *Salmonella* phage GJL01 (GenBank Acc. No. KY657202.1), 97.97% nucleotide similarity with *Escherichia* phage vB_EcoS_011D2 (GenBank Acc. No. MT478992.1), 96.86% nucleotide similarity with *Escherichia* phage DanielBernoulli strain Bas08 (GenBank Acc. No. MZ501059.1), and 96.94% nucleotide similarity with *Citrobacter* phage vB_CfrD_Brooksby (GenBank Acc. No. OL539443.1) (Figure 12, Appendix A).

*Salmonella* phage strain KKP 3830 similarities on a 2D scatter plot computationally predicted and rendered through PhageAI v.1.0.2. software are shown in the Appendix A (Appendix A).

### 2.8. Influence of Selected Factors on the Preservation of the Activity of Phages

In this study, the lytic activity of isolated phages exposed to a wide range of temperatures (from −20 to 80 °C), active acidity (pH from 2 to 12), and time of UV exposure (0, 5, 10, 30, and 60 min) have been evaluated. The lytic activity was expressed as phage titer (in log PFU mL^−1^).

Temperature is the crucial factor for bacteriophages’ viability and activity in the environment [69]. It plays a crucial role in the attachment, penetration, and amplification of phage particles in their bacterial host cells [70]. The effect of a wide range of temperatures on the activity of phages is presented in Figure 13.

Both phages retained their activity at freezing and refrigeration temperatures. *Salmonella* phage strain KKP 3829 and *Salmonella* phage strain KKP 3830 were stable up to 50 °C and 60 °C, respectively. For salmophage KKP 3829, the temperature of 60 °C significantly reduced (*p* ≤ 0.05) their activity by 90% compared to lower temperatures, while at higher temperatures (70 °C and 80 °C), the phages were inactivated. In the case of *Salmonella* phage strain KKP 3830, at 70 °C, approximately 99% of the virus particles were inactivated (*p* ≤ 0.05). A higher temperature (80 °C), as in the case of salmophage KKP 3829, had an inactivating effect on them. The effect of temperature on phage activity is commonly examined during the characterization of bacteriophages. In the research of Inbaraj et al. [71], bacteriophage vB_SenS_Ib_psk2 against MDR *S*. Kentucky retained optimal activity from 20 °C to 42 °C. In another study, there was no significant loss of salmophage LPST153 count between 30 °C and 60 °C, while the optimum temperature for phage stability was between 30 °C and 50 °C. However, the phage count was reduced by approximately 75% at 70 °C, indicating that salmophage LPST153 exhibited moderate heat resistance [46]. Another phage, salmophage BIS20, produced maximum titer at 37 °C; however, it was still able to infect at temperatures up to 45 °C, albeit with a significant drop in titer [21]. Heat stability tests revealed that salmophage ZCSE9 was completely inactive at 90 °C with a significant reduction in the activity of the phage at 85 °C. No significant reduction occurred in the stability of the phage after incubation for one hour at the temperature range of −20 °C to 80 °C [47].

The effect of a wide range of active acidity on the activity of phages is presented in Figure 14. Both phages were stable over a wide range of active acidity (pH 3 to 11). Extreme pH values (2 and 12) completely inactivated phage virions (*p* ≤ 0.05). Similar results were obtained by the authors in other studies on phages targeting *Salmonella*. Salmophage ZCSE9 maintained high activity in the range of pH 4 to 11, with the highest titer observed at pH 7. Its activity rapidly decreased at both pH 3 and pH 12 and was inactive at pH 2 [47]. The highest titer of salmophage BIS20 was observed at pH 7, whereas a significant reduction in titer was observed at pH 9 after 1 h of incubation. No plaque zones were observed at pH 2, 5, and 12 [72]. The pH stability test of the salmophage LPST153 showed that it was highly stable with pH ranging from 4 to 12. However, this phage was completely abolished under strong acids or strong alkalis (pH < 4 or pH > 12) [46].

Figure 15 shows the effect of UV exposure time on phages’ activity. The exposure of all phages to UV radiation significantly decreased their lytic activity in proportion to the exposure time (*p* ≤ 0.05). It is worth noting that an hour’s exposure to UV radiation is not an effective method of the complete inactivation of phage virion particles. 

The stability of phages exposed to UV radiation is much less frequently determined. In the study by Wang et al. [73], the broad host-spectrum virulent salmophages, fmb-p1, was stable after 30 min of UV exposure, although a 1-log decrease in phage titer after 1 h of UV exposure was noted. Completely different results were presented in the study by Kim et al. [74], in which the lytic salmophage, SS3e, lost 50% of its viability after 1 min, 90% of the phage particles were inactivated (one log unit) after 5 min, and after 15 and 20 min of exposure to UV radiation, the phage particles were undetectable. 

### 2.9. Application of the Phage Cocktail to the Analyzed RTE Food Products

In the last stage of this research, the effect of the phage cocktail on the growth of *Salmonella* rods in minimally processed juices was determined. Two juices were selected for application tests: carrot–mango–apple juice preserved with high hydrostatic pressure (HHP; 600 MPa) and raw carrot–apple juice. A multiply of infection of 1.0 was used. Figure 16 shows changes in *Salmonella* counts in HHP-preserved and phage-treated juice during storage at 4 °C.

Considering the active acidity (pH) of juices during the seven-day refrigerated storage, no significant changes were observed. The application of either phage individually or the phage cocktail to HHP-preserved juice significantly reduced the growth of *Salmonella*. The exception was the addition of *Salmonella* phage strain KKP 3830 to juice infected with *Salmonella enterica* subsp. *enterica* serovar 6,8:l,-:1,7 strain KKP 1762, where no significant reduction in the pathogen was demonstrated (Figure 16, Table 8).

The level of *Salmonella enterica* subsp. *enterica* serovar 6,8:l,-:1,7 strain KKP 1762 contamination was similar to the control samples, which is related to the lack of activity of *Salmonella* phage strain KKP 3830 against this pathogen (see Table 3). Each of the phages applied individually reduced the growth of the pathogen to a different extent. The growth of pathogens was limited the most by the phage for which the given bacterial strain was the target host (i.e., the growth of *Salmonella enterica* subsp. *enterica* serovar 6,8:l,-:1,7 strain KKP 1762 was limited the most effectively by *Salmonella* phage strain KKP 3829, while *Salmonella* Typhimurium strain KKP 3080 was limited the most by *Salmonella* phage strain KKP 3830).

The strongest inhibitory effect on the growth of the pathogen was obtained with the use of a cocktail composed of all four bacteriophages. Concerning the number of *Salmonella enterica* subsp. *enterica* serovar 6,8:l,-:1,7 strain KKP 1762 in the juices treated with *Salmonella* phage strain KKP 3829 or the phage cocktail, the bacteria count was similar at the end of the refrigerated storage. The use of a phage cocktail reduced the growth of both strains of bacterial pathogens at the end of refrigerated storage by about a one log unit (90%) compared to non-phage-treated (control) samples.

The second tested product was raw carrot–apple juice. The effect of the phage cocktail on the inhibition of the growth of *Salmonella* count was determined both during refrigerated storage (4 °C) and at 20 °C (Figure 17).

The addition of a phage cocktail to juice infected with *Salmonella enterica* subsp. *enterica* serovar 6,8:l,-:1,7 strain KKP 1762 and stored at 4 °C over 24 h significantly reduced (*p* ≤ 0.0001) the number of pathogenic bacteria. In the case of the second *Salmonella* strain (*S*. Typhimurium strain KKP 3080), a significant reduction (*p* ≤ 0.05) occurred after 48 h. In the juices infected with *Salmonella* and stored at 20 °C, after 24 h in the control samples, an increase in the rods by about one log unit compared to the initial number was observed, which can be explained by the possibility of *Salmonella* growth at room temperature (20 °C). In phage-treated samples stored at 20 °C, there was a significant reduction (*p* ≤ 0.0001) in *Salmonella* rods after 24 h compared to control samples. During the further storage of the juices at 20 °C (2 days and more), as in the case of the seven-day storage of the juice at 4 °C, no *Salmonella* was found in the cultures (LOD = 1 CFU mL^−1^, LOQ = 4 CFU mL^−1^) both in the control samples and in the phage-treated samples. The inhibition of *Salmonella* growth can be caused by rapid changes in the pH of the food matrix (Figure 17; right side of the graphs). These changes in pH are probably due to the use of raw juice, which was not a sterile product and could contain autochthonous strains, e.g., acidifying bacteria. *Salmonella* grows around pH 4–9, so lowering the pH to 3.9 and below may have stunted its growth.

The use of a four-phage cocktail of different bacteriophages did not allow to achieve a complete reduction in *Salmonella* in the food matrix, but there was a significant inhibition of the growth of this pathogen. Although *Salmonella enterica* subsp. *enterica* serovar 6,8:l,-:1,7 strain KKP 1762 was completely inhibited in the in vitro study (see Section 2.5) at MOI = 1.0, no such effect was obtained in the in vivo study. Therefore, in the future, the use of a higher-titer phage cocktail may be the right approach. In addition, *Salmonella* phage strain KKP 3830 showed no activity against *S*. I (6,8:l,-:1,7) strain 1762 (see Table 3), which resulted in a depletion of the biopreparation. On the other hand, it can be observed that after 24 h at 20 °C, there was a significant reduction in *S*. I (6,8:l,-:1,7) strain 1762 (by about 99.999% compared to the control; Figure 17B), which is also confirmed in the in vitro study (Figure 6A). Finally, the pH of the food matrix also affected the activity of the biopreparation. From the in vitro study, we know that the bacteriophages were stable for 1 h at low active acidity (pH 3–4), but no long-term stability studies were performed. Further studies will be performed in parallel, both in sterile and raw food matrix intentionally infected with a bacterial pathogen.

The concept of using phages for food preservation was developed many years ago. In countries outside the European Union (e.g., USA, Canada, and New Zealand), phage preparations are used commercially in the food industry. Scientists in European centers are also conducting extensive research on the use of phages in food biopreservation. The effectiveness of phage biopreparations is assessed in various food matrices. Islam et al. [75] evaluated the effectiveness of a phage cocktail in a food matrix of animal origin. Application of a three-phage cocktail with a wide range of *Salmonella* bacterial hosts resulted in a significant decrease (by approximately 99.9%) in pathogenic bacteria in milk and chicken breast. In addition, the phage-cocktail application caused a significant reduction in bacterial biofilm, both in vitro and on stainless steel surfaces. In another study, salmophage LPST94 at MOI 1000 and 10000 reduced *Salmonella* by 99.9% in milk, apple juice, chicken breast, and lettuce [76]. Bao et al. [77] assessed the effect of the addition of a phage cocktail on the growth of *Salmonella* in milk and Chinese cabbage. A significant reduction in the number of pathogenic bacteria was observed (1.5–4.0 log CFU sample^−1^), with the phage cocktail being more effective than a single-phage application.

In conclusion, the effectiveness of the phage cocktails in the reduction (or complete elimination) in pathogenic bacteria in the food environment was confirmed.

## 3. Materials and Methods

### 3.1. Bacterial Host Strains

For the bacteriophages’ isolation, two strains of *Salmonella enterica* subsp. *enterica* were used, i.e., *S*.I (6.8:l,-:1,7) strain KKP 1762 and *S*. Typhimurium strain KKP 3080. GenXone SA (Złotniki, Poland) was commissioned to sequence all the bacterial genomes. For this purpose, bacterial genetic material was isolated using the Genomic Mini AX Bacteria kit (A&A Biotechnology, Gdynia, Poland), according to the manufacturer’s protocol. DNA libraries were prepared using Rapid Barcoding Kit reagents (Oxford Nanopore Technologies, Oxford, UK), according to the manufacturer’s protocol. A sequencing depth of at least 50× genome coverage was assumed. NGS sequencing was performed in the nanopore technology on the GridION X5 sequencing device (Oxford Nanopore Technologies, Oxford, UK) under the control of MinKnow v22.10.5. Bases were called with Guppy v6.3.8 Basecaller (Oxford Nanopore Technologies, Oxford, UK), followed by barcode demultiplexing, also using Guppy Barcoder v6.3.8 (Oxford Nanopore Technologies, Oxford, UK), generating a .fastq file for each barcode. De novo assembly of genomes was performed in Flye v2.8.1 software [78] and annotation of bacterial genomes in Prokka 1.14.6 v1.1.0 software [31]. Proksee software [32] was used to visualize bacterial genomes. The RGI 5.2.1 v1.1.1 software was used to search for antibiotic resistance genes (ARGs) in bacterial genomes [79]. Alien_hunter 1.7 v1.1.0 software was used to predict putative horizontal gene transfer (HGT) events [80]. Mobile genetic elements (MGEs) were searched in mobileOG-db (beatrix-1.6) v1.1.2 software [81]. CRISPR arrays and their associated Cas proteins were analyzed in CRISPRCas-Finder 4.2.20 v1.1.0 software [82]. VirSorter 2.2.4 v1.1.1 [83] and Phigaro 2.3.0 v1.0.0 [84] software were used to detect and characterize prophage regions in bacterial genomes. The genomes of the host bacterial strains were deposited in the GenBank database.

### 3.2. Bacteriophage Isolation, Purification, and Propagation

A total of 25 mL of municipal sewage (Czajka Wastewater Treatment Plant, Warsaw, Poland) was centrifuged at 10,000× *g* (20 °C for 10 min; ultracentrifuge Sorvall LYNX 6000, Thermo Fisher Scientific, Watertown, MA, USA) to separate organic and mineral particles from bacteria and potential bacteriophages. The supernatant was filtered using a syringe filter with a membrane pore diameter of 0.22 μm (Minisart^®^ NML Cellulose Acetate; Sartorius, Goettingen, Germany). Then, 20 mL of the filtrated supernatant containing bacteriophages from the sewage was transferred to 20 mL of double-concentrated Luria–Bertani broth (BTL, Lodz, Poland). The culture medium with bacteriophages was inoculated with 1 mL of an overnight culture of a bacterial strain on a Luria–Bertani broth and incubated at 37 °C for 24 h. Afterward, the culture was centrifuged at 8000× *g* for 10 min (ultracentrifuge Sorvall LYNX 6000, Thermo Fisher Scientific, Watertown, MA, USA) to separate bacteria from the proliferated bacteriophages. The supernatant was filtered using a syringe filter with a membrane pore diameter of 0.45 μm (Minisart^®^ NML Cellulose Acetate; Sartorius, Goettingen, Germany; according to Mirzaei and Nilsson procedure [85] with modification) and freeze-stored (–80 °C) with a 20% addition of glycerol.

The phage concentration was determined in each lysate using the double-layer agar-plate method [86] in triplicate. For this purpose, a standard dilution series of lysates were prepared. Then, for every 500 µL of lysate, 100 µL of the bacterial host was added. After a 15 min incubation, 4 mL of soft agar (Luria–Bertani with 0.75% agar) was added and poured onto a plate with a layer of nutrient agar (BTL, Lodz, Poland). After 24 h incubation at 37 °C, plaques which formed on the bacterial lawn were computed using the formula below, considering the dilution factor and expressed as phage titer (P_T_—plaque forming units in 1 mL of lysate, PFU mL^−1^):(1)$$P_{T}=\frac{C}{R}*2$$
where: C—number of plaques on Petri dish, (PFU); R—phage lysate dilution factor, (–); 2—factor of result conversion to 1 milliliter, (mL^−1^). 

Single-phage (transparent with or without ‘halo’ zone or turbid) plaques were cut with a scalpel and purified in saline magnesium (SM) buffer according to the method proposed by Mirzaei and Nilsson [85]. Purification was performed in four rounds of single-plaque passage to ensure that the isolate represented the clonal phage population. In addition, after filtering through a syringe filter with a membrane pore diameter of 0.45 µm, each lysate was inoculated to check for possible contamination with bacterial cells. 

### 3.3. Spot Test to Determine the Range of Bacterial Hosts for the Tested Phages

In total, 81 diverse bacterial strains were used to examine the phage lytic range, which consisted of one group of 54 *Salmonella enterica* strains and a group of 28 non-*Salmonella enterica* strains. The bacterial strains have been isolated and deposited in the Culture Collection of Industrial Microorganisms—Microbiological Resources Center (IAFB). The taxonomic identification of most of the bacteria used was determined during previous studies or in this study, in accordance with the procedure described in our articles [14,87]. Moreover, the *Salmonella* strains were sent to the National *Salmonella* Center (Medical University of Gdańsk, Gdansk, Poland) for serotyping. For this purpose, they were characterized biochemically and serotyped according to standard techniques. The antigenic factors were identified by *Salmonella*-specific rabbit antisera. The White–Kauffmann–Le Minor scheme was used to name the serovars.

The ability of phages to infect bacteria with different strains was determined by spot test [46]. In brief, 5 μL lysates from each phage (phage titer ~10^7^ PFU mL^−1^) were spotted onto bacteria lawns, which were separately poured with 81 bacterial strains, in which the strains were propagated on nutrient agar plates. Four milliliters of 0.75% soft agar overlay were applied to enable transparent spots/plaques development. The lytic range activities of all bacterial strains were determined at 37 °C with 24 h incubation in triplicate. After the incubation, transparent spots/plaques on any bacterial lawns were recorded as corresponding phage sensitive, according to the signs: “++”—transparent plaques; “+”—turbid plaques; “–“—no plaques (non-susceptible bacterial strain).

### 3.4. One-Step Growth

One-step growth curve experiments have been accomplished to assess the latent period, rise period (burst time), and burst size, according to a method described by Islam et al. [46] and Shakeri et al. [88], with our modification. Firstly, host bacterial strains were allowed to grow to mid-log phase, and then, 5 mL of bacterial culture (1.0 × 10^7^ CFU mL^−1^) and 5 mL of phage lysate (1.0 × 10^6^ PFU mL^−1^) were added to 40 mL of fresh Luria–Bertani broth to achieve a multiplicity of infection (MOI) of 0.1. The phage-host mixture was incubated for 10 min at 37 °C and was subsequently centrifuged at 7000× *g* for 2 min at 4 °C (ultracentrifuge Sorvall LYNX 6000, Thermo Fisher Scientific, Watertown, MA, USA). Afterwards, the supernatant with excess free phages was discarded and the bacterial pellet was washed twice with Luria–Bertani broth. Then, the pellet was suspended in 50 mL of fresh Luria–Bertani broth and incubated at 37 °C for the duration of the experiment. Samples (2 mL) were withdrawn at 5 min intervals by 2 h. Each sample was centrifuged for 1 min at 10,000 rpm (centrifuge MiniSpin^®^ plus, Eppendorf, Hamburg, Germany) immediately after collection, and the supernatant was transferred to a new tube. Experiments were performed in triplicate by the double-layer agar plate method. The latent period was defined as the time interval between absorption and the beginning of the first burst. The burst size was summarized as the ratio of the final number of phage particles and the initial number of host bacteria at the beginning of the experiment [73,89].

### 3.5. Phage Adsorption to Host Bacterial Cells

To determine the phage adsorption constant and assess the phage adsorption kinetics to host bacterial cells, the procedure described by Bagińska et al. [90] was used. In brief, the same volumes of phage lysate and 4 h host bacterial suspension (20 mL of each) were mixed at MOI = 0.1 (10^6^ PFU mL^−1^ and 10^7^ CFU mL^−1^, respectively). Immediately after mixing and after specific incubation intervals (i.e., 0, 1, 2, 3, 4, 5, 7.5, 10, 12.5, 15, 17.5, and 20 min) at 37 °C, 2 mL of the sample was taken and filtered (syringe filter with a membrane pore diameter of 0.45 µm). The number of free, non-adsorbed phage particles in the supernatant was determined using the double-layer agar plate method in triplicate. The number of phages immediately after mixing with the host bacterial suspension was considered to be 100% of free phage particles, and other titers were compared to this value. The adsorption rate constants (*k*) were determined according to the following formula:(2)$$k=\frac{2.3}{\mathrm{Bt}}\log\frac{P0}{P}$$
where: k—adsorption rate constant (mL min^−1^); B—concentration of host bacterial cells; *t*—time interval in which the phage titer falls from P0 (original titer) to P (final titer). 

### 3.6. Changes in the Growth Kinetics of Bacterial Hosts after Phage Infection

Growth curves were made for each of the bacterial strains (data unpublished in this study). For this purpose, the harvested culture was inoculated on PCA medium every hour, incubated at 37 °C for 24 h, and the optical density was measured simultaneously (DU^®^ 640 Spectrophotometer, Beckman Instruments, Inc., Fullerton, CA, USA). The dependence of the optical density on the number of bacterial cells was determined (performed in triplicate). 

Once phage titer was determined and bacterial host growth curves were plotted, bacteria growth kinetics was measured using a Bioscreen C Pro automated growth analyzer (Yo AB Ltd., Growth Curves, Helsinki, Finland). Bacteria proliferated in the Luria–Bertani broth. The culture was diluted at a ratio of 1:100 in a fresh culture medium with the addition of CaCl_2_ and MgSO_4_, both having final concentrations of 0.01 M. To ensure the optimal value of the multiplicity of infection (MOI) coefficient, flasks with the new culture were incubated at a temperature of 37 °C with continuous shaking until the desired optical density depended on the phage titer. Then, 180 µL of each culture was pipetted into multi-well plates and incubated in a Bioscreen C Pro at 37 °C until optical density increased by OD_600_ ~0.1, compared to the control medium. Phage lysates were prepared so that the value of the MOI coefficient reached 1000, 100, 10, 1.0, 0.1, 0.01, 0.001, and 0.0001, respectively. Then, 20 µL of respective phage lysates were added to wells, left at 20 °C for 20 min to allow the phages to adsorb to the host cell surface, and incubated at 37 °C for 48 h. The apparatus automatically measured the optical density every 15 min at a wide band of wavelengths ranging from 400 to 600 nm, with 15 s shaking preceding each readout. The test was performed in 10 replicates for each strain and infection rate.

Next, coefficients of the specific growth rate (µ) were computed for each strain using the following formula:$$µ=\frac{\ln{\mathrm{OD}}_{\max}-\ln{\mathrm{OD}}_{\min}}{t}$$
where: ln OD_max_—natural logarithm of the maximal value of the optical density of the culture during the exponential growth phase; ln OD_min_—natural logarithm of the minimal value of the optical density of the culture during the exponential growth phase; t—duration of the exponential growth phase, (h).

### 3.7. Determination of Morphological Features of Phages

Transmission electron microscopy (TEM) was used for determining morphological features of the isolated bacteriophages and classifying them into respective families. Propagated phage lysates were centrifuged at 4 °C and 14,500 rpm for 40 min (centrifuge MiniSpin^®^ plus, Eppendorf, Hamburg, Germany). The excess culture medium was removed, and the pellet was suspended in 2 mL of 100 mM cold ammonium acetate (filtered through a syringe filter with a membrane pore diameter of 0.22 µm). The precipitate was disintegrated with a tip and centrifuged again. The whole procedure was repeated four times. After centrifugation, the precipitate was flushed from the Eppendorf tube wall with 50 µL of ammonium acetate according to Ackermann’s procedure [91] with modification. An amount of 2 µL of the phage suspension in ammonium acetate was coated onto carbon-sputtered copper–wolfram mesh grids. After drying, the specimen was stained in 2% uranyl acetate solution for 1 min. Samples were dried for 12 h at ambient temperature under sterile conditions (according to Ackermann [92], Amarillas et al. [93], Mahmoud et al. [52] procedures with modification) and visualized under JEM-1400 PLUS transmission electron microscopy (Japan Electron Optics Laboratory Co., Ltd., Tokyo, Japan) in 60,000–100,000× magnification, at a voltage of 80 kV [94]. The images were dimensioned in SightX v.2.1 software (Japan Electron Optics Laboratory Co., Ltd.).

### 3.8. Extraction of Bacteriophage Genomic DNA

Phage lysates were concentrated by ultracentrifugation. For this purpose, 40 mL of phage lysate was transferred to dedicated bottles, and 8 mL of precipitation solution (PEG–NaCl; 20% PEG 8000 2.5 M NaCl) was added. The mixture was incubated on ice overnight. The next day, the lysates were ultracentrifuged at 4 °C for 1 h at 27,000 rpm (ultracentrifuge Sorvall LYNX 6000, Thermo Fisher Scientific, Watertown, MA, USA). Bacteriophage genomic DNA was isolated using PureLink^TM^ RNA/DNA Mini Kit (Thermo Fisher Scientific Inc., Carlsbad, CA, USA) according to the manufacturer’s protocol with our modifications. An amount of 400 µL lysis buffer (containing 5.6 µg carrier RNA) was added to the phage pellet and vortexed for 15 s. Then, 50 µL of proteinase K was added and incubated at 56 °C for 1 h with shaking at 900 rpm (ThermoMicer C, Eppendorf, Hamburg, Germany). After incubation, the tubes were briefly centrifuged to remove any drops from the inside of the lids. An amount of 250 µL of 100% ice-cold ethanol (molecular biology grade) was then added to the samples, vortexed for 15 s, and incubated for 5 min at room temperature (20 °C). After incubation, the tubes were briefly centrifuged to remove any drops from the inside of the lids. A total of 675 µL samples were transferred to the viral spin column and centrifuged for 1 min at 10,000 rpm (centrifuge MiniSpin^®^ plus, Eppendorf, Hamburg, Germany). Columns were transferred to new wash tubes, and 500 µL of wash buffer was added and centrifuged for 1 min at 10,000 rpm. The procedure was repeated 3 times, with the last centrifugation for 3 min at 14,500 rpm. To elute the genetic material, the viral spin column was transferred to new tubes and 20 µL of RNase-free water was added. After a 1 min incubation, the samples were centrifuged twice for 1 min at 14,500 rpm. DNA purity was measured by the Nanodrop ND-1000 Spectrophotometer (Thermo Fisher Scientific, Watertown, MA, USA), and DNA concentration was quantified by a Qubit 4.0 Fluorometer using the Qubit dsDNA BR Assay Kit (Invitrogen, Carlsbad, CA, USA). DNA samples were stored at 4 °C until further processing for whole-genome sequencing (WGS) analysis. 

### 3.9. Phage Genome Sequencing and Bioinformatics Analysis

GenXone SA (Złotniki, Poland) was commissioned to sequence all the phage genomes. DNA libraries were prepared using Rapid Barcoding Kit reagents (Oxford Nanopore Technologies, Oxford, UK) according to the manufacturer’s protocol. A sequencing depth of at least 50x genome coverage was assumed. NGS sequencing was performed by nanopore technology on the GridION X5 sequencing device (Oxford Nanopore Technologies, Oxford, UK) under the control of MinKnow v22.10.5. Bases were called with Guppy v6.3.8 Basecaller (Oxford Nanopore Technologies, Oxford, UK), followed by barcode demultiplexing, also using Guppy Barcoder v6.3.8 (Oxford Nanopore Technologies, Oxford, UK), generating a .fastq file for each barcode. De novo assembly of genomes was performed in Flye v2.8.1 software [78] and annotation of phage genomes in Phanotate v1.5.0 [95] and PhaGAA software [96]. Proksee software [31] was used to visualize phage genomes. Viral proteomic trees of phage genomes were calculated by BIONJ based on genomic distance matrixes and mid-point rooted and were represented in the circular view. Branch lengths were log-scaled. The sequence and taxonomic data were based on Virus–Host DB [57]. The trees were generated using the ViPTree server [58]. Genome sequence comparison of newly isolated phages with five other related phage genomes exhibiting co-linearity were detected by TBLASTX using FastANI 1.3.3 software [68] and Proksee software [32]. The phylogenetic relatedness of our phages against the most common phage hits was constructed using the neighbour-joining method in the NCBI database [97]. Phage similarity was computationally predicted on a 2D scatter plot and rendered through PhageAI v.1.0.2 software [98]. The phage genomes were deposited in the GenBank database.

### 3.10. Influence of Selected Factors on the Preservation of the Activity of Phages

This stage of the experiment aimed to determine the activity of bacteriophages exposed to a wide range of temperatures, pH values, and time of UV exposure. To determine the activity of phage lysates at various temperatures (–20 °C, 4 °C, 20 °C, 30 °C, 40 °C, 50 °C, 60 °C, 70 °C, and 80 °C), 100 µL of the suspension was added to 9.9 mL of physiological saline with pH 7.0. The mixture was held for 1 h at specified temperatures [99]. To determine the activity of phage lysates at various active acidity, 100 µL of the phage lysate were added to test tubes containing 9.9 mL of sterile physiological saline (0.85% NaCl) with a fixed active acidity (pH in the range from 2 to 12). The mixture was held at 20 °C for 1 h. To determine the effect of UV radiation, phage lysates were exposed to UV radiation for 0, 5, 10, 30, and 60 min. The experiments were carried out in three independent replicates. After the stage of phage exposure to chemical or physical factors, a microdilution method was employed to prepare lysates with varying titers. Phage concentration as a phage titer was determined by the double-layer plate-agar method.

### 3.11. Application of the Phage Cocktail to the Analyzed RTE Food Products

The effect of the phages on the limitation of the growth of *Salmonella* rods in minimally processed ready-to-eat juices was determined. Two juices were selected for application tests: HHP-preserved carrot–mango–apple and raw carrot–apple juices. 

#### 3.11.1. HHP-Preserved Carrot–Mango–Apple Juice

The food matrix was commercially available ready-to-eat HHP-treated (600 MPa) carrot–mango–apple juice (10% carrot/20% mango/70% apple). Juice samples were infected with a specific bacterial strain, i.e., *S*. I (6,8:l,-:1,7) strain KKP 1762 or *S*. Typhimurium strain KKP 3080, to a final count of 10^5^ PFU mL^−1^. Single phages or a phage cocktail were applied to the juices. The phage cocktail was a mixture of four bacteriophages (two characterized in this study, and two others, i.e., vB_Sen-IAFB3822 (*Salmonella* phage strain KKP 3822; GenBank accession number: OQ674104), and vB_Sen-IAFB3831 (*Salmonella* phage strain KKP 3831; GenBank accession number: OQ674103) with known titer lysates. Before infecting the products, the single phages or phage cocktail were filtered through a 0.45 μm cellulose syringe filter. The titer of the prepared phage cocktail equaled 6.9 × 10^7^ PFU mL^−1^ and that was the average number of all phages in the biopreparation. The single phage biopreparations or phage cocktail were added to the juice to obtain an MOI of 1.0. The samples were stored at 4 °C. The changes in the count of *Salmonella* during refrigerate storage were examined in triplicate at the 0th, 24th, 48th, 72nd, 120th, and 168th hours of the experiment. Brilliance™ *Salmonella* Agar (Oxoid, Thermo Scientific, Basingstoke, UK) was used for culturing. Simultaneously, the control samples (without the addition of the single phages or phage cocktail) were analyzed. Moreover, pH changes were monitored for each sample.

#### 3.11.2. Raw Carrot–Apple Juice

The second food matrix was commercially available ready-to-eat raw carrot–apple juice (90% carrot/10% apple). Juice samples were infected with a specific bacterial strain, i.e., *S*. I (6,8:l,-:1,7) strain KKP 1762 or *S*. Typhimurium strain KKP 3080, to a final count of 10^5^ PFU mL^−1^. Only the phage cocktail, which was prepared as before (see in Section 3.11.1.), was applied to the juices. The phage biopreparation was added to the juice to obtain an MOI of 1.0. The samples were stored at 4 °C or 20 °C. The changes in the count of *Salmonella* during storage were examined in triplicate at the 0th, 24th, 48th, 72nd, 120th, and 168th hours of the experiment. Brilliance™ *Salmonella* Agar (Oxoid, Thermo Scientific, Basingstoke, UK) was used as a culture medium. Simultaneously, the control samples (without the addition of the phage cocktail) were analyzed. Moreover, pH changes were monitored for each sample.

### 3.12. Statistical Analysis

All the experiments were repeated at least three times. All data presented graphically or in tables were subjected to statistical analyses performed using Graph Prism v9.4.1 (GraphPad Software Inc., San Diego, CA, USA) unless otherwise stated. The effect of phages at different MOI rates on the growth kinetics of bacterial hosts was analyzed using a two-way analysis of variance (ANOVA) followed by multiple comparison Dunnett’s test with a 95% confidence interval (α = 0.05). One-way ANOVA followed by Tukey’s test with a 95% confidence interval (α = 0.05) was used to analyze the impact of selected physical or chemical factors on phages’ activity. Multiple t-test followed by a two-stage step-up method by Benjamini, Krieger, and Yekutieli, with a correction for multiple comparisons using the Holm–Sidak method (α = 0.05) was used to analyze the practical application of the phage cocktail to food matrices. 

## 4. Conclusions

The search for and the possibility of using bacteriophages for food biocontrol is the subject of interest for a growing group of scientists around the world, including in the European Union, where the use of phage biopreparations in food production is not allowed. Phage biopreparations targeting various bacterial pathogens, most often found in food, including *Salmonella* sp., *Listeria monocytogenes*, and *Campylobacter jejuni*, are being developed and tested in vitro and in vivo. Particular attention has recently been paid to the problem of the occurrence of bacteria resistant to antibiotics, the source of which may be food.

Each area/region is characterized by its own indigenous microbiota, specific to a given area. It is the same with bacteria in food—a group/specific strains of bacteria dominate in each area. In the case of *Salmonella*, it is even serovars, some of which occur in a given country, and the occurrence of others has never been recorded. Therefore, an attempt to isolate bacteriophages targeted at autochthonous MDR *Salmonella* strains, and then to characterize them at the morphological, genomic, and application level, was also undertaken by our team from the Prof. Waclaw Dabrowski Institute of Agricultural and Food Biotechnology—State Research Institute. In this article, we have presented two out of numerous different bacteriophages isolated against MDR *Salmonella* strains. Genomic analyses indicate that both newly isolated salmophages, *Salmonella* phage strain KKP 3829 and *Salmonella* phage strain KKP 3830, are strictly lytic viruses lacking other virulence factors. In vitro studies significantly reduced the growth of bacterial hosts, regardless of the infection factor used. Both phages show a wide tolerance to environmental factors, including temperature, pH, or UV radiation, which may enable their use in combination with other chemical or physical means of food protection or disinfection of production lines in the food industry. Finally, in in vivo studies, the applied phage cocktail effectively limited the growth of pathogenic bacteria. Our research shows that tested phages have potential effectiveness as a biocontrol agent of *Salmonella* in foods.

In further research, the effectiveness of our virulent phages in other food matrices and the effectiveness of the phage cocktail in combination with physical, chemical, or physicochemical techniques for disinfecting the surface of production lines, including the eradication of bacterial biofilms, will be addressed.

## Figures and Tables

**Figure 4 ijms-24-10134-f004:**
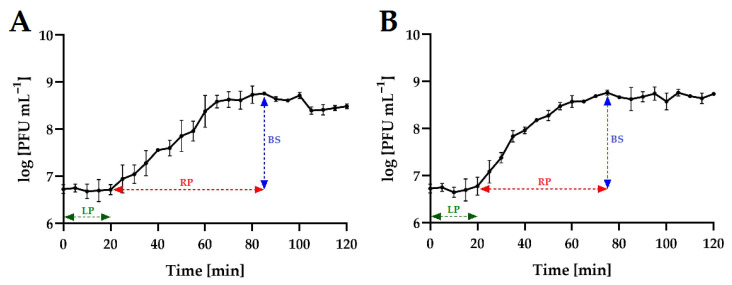
One-step growth curve of *Salmonella* phages at MOI = 0.1. Points represent an average; error bars represent the standard deviation (±SD) of the mean phage titers for three repetitions (*n* = 3): (**A**)—*Salmonella* phage strain KKP 3829; (**B**)—*Salmonella* phage strain KKP 3830. Symbols in charts: LP—latent period; RP—rise period (burst time); BS—burst size.

**Figure 5 ijms-24-10134-f005:**
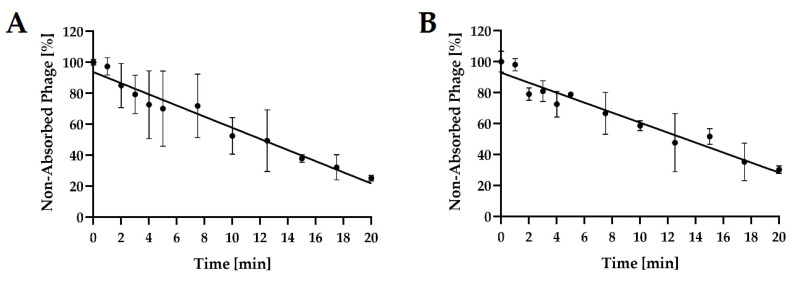
Kinetic of *Salmonella* phages adsorption at MOI = 0.1. Points represent an average; error bars represent the standard deviation (±SD) of the mean phage titers for three repetitions (*n* = 3): (**A**)—*Salmonella* phage strain KKP 3829; (**B**)—*Salmonella* phage strain KKP 3830.

**Figure 6 ijms-24-10134-f006:**
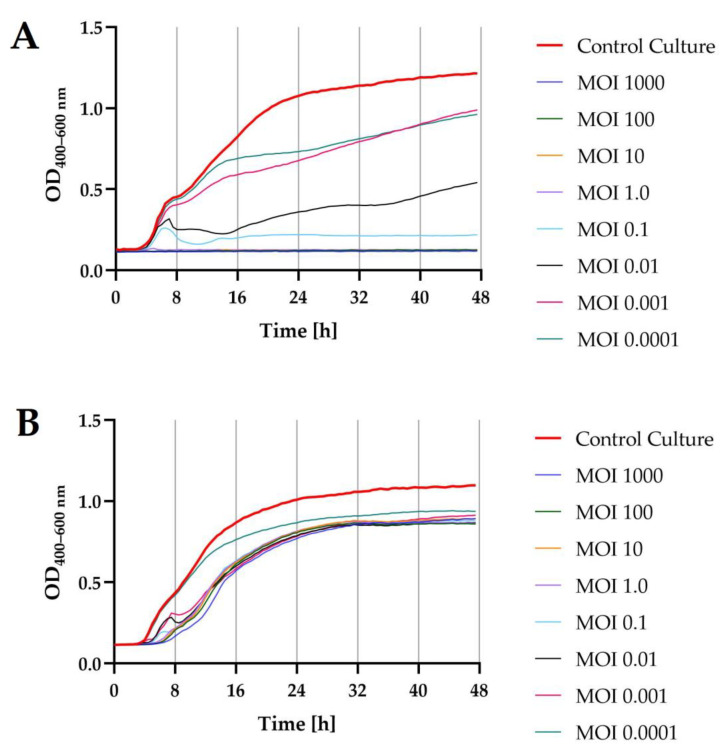
Growth curves of the bacterial host strains (*n* = 10) treated with phages at different infection coefficients of MOIs (1000; 100; 10; 1.0; 0.1; 0.01; 0.001; and 0.0001) compared to the control culture (red bold line): (**A**)—*S*. I (6,8:l,-:1,7) strain KKP 1762 + *Salmonella* phage strain KKP 3829; (**B**)—*S*. Typhimurium strain KKP 3080 + *Salmonella* phage strain KKP 3830.

**Figure 7 ijms-24-10134-f007:**
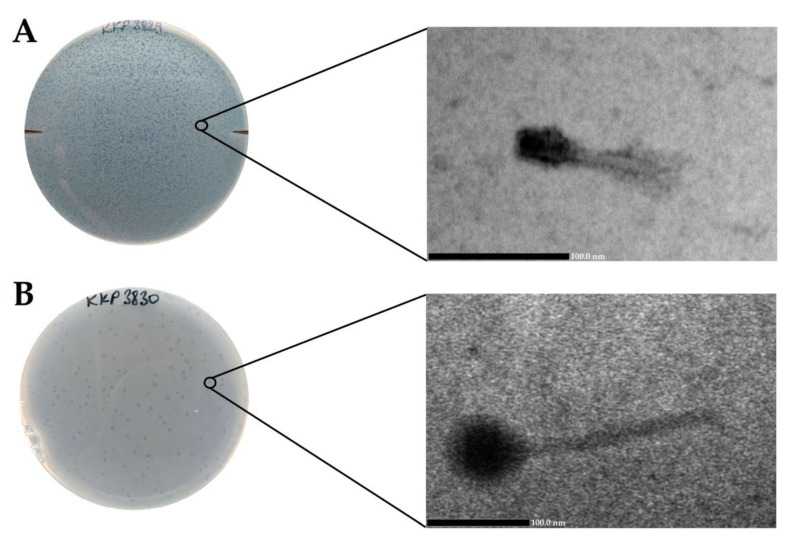
Morphology of plaques photographed on agar plates (left side) and electron micrographs from the TEM (right side) showing the morphology of bacteriophages: (**A**)—*Salmonella* phage strain KKP 3829 (100,000×); (**B**)—*Salmonella* phage strain KKP 3830 (60,000×).

**Figure 8 ijms-24-10134-f008:**
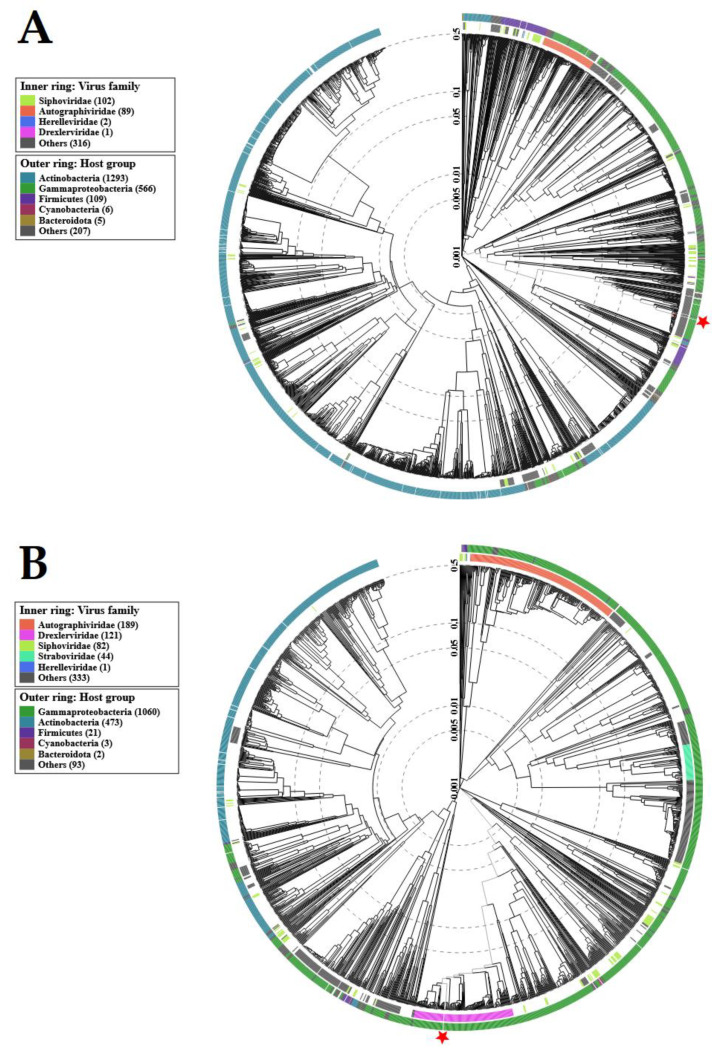
Viral proteomic trees of: (**A**)—*Salmonella* phage strain KKP 3829; (**B**)—*Salmonella* phage strain KKP 3830, and other phage genomes are represented in the circular view. The branch which represented studied phages is marked by an asterisk. Color rings indicate virus families (inner rings) and host groups (at a level of phylum; outer rings). These trees were calculated by BIONJ based on genomic distance matrixes and mid-point rooted. Branch lengths are log-scaled. The sequence and taxonomic data were based on Virus–Host DB [57]. The trees shown were generated using the ViPTree server [58].

**Figure 9 ijms-24-10134-f009:**
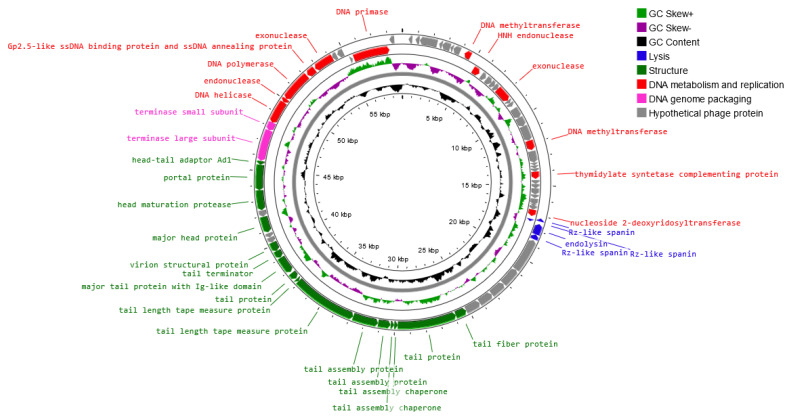
Map of the genome organization of *Salmonella* phage strain KKP 3829 generated using the Proksee software [32].

**Figure 10 ijms-24-10134-f010:**
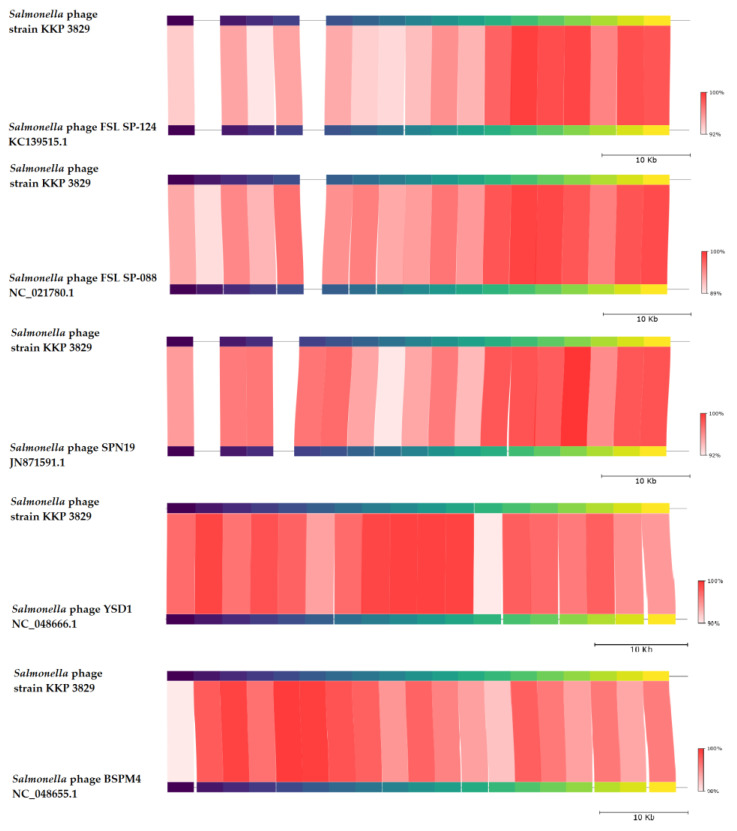
Genome sequence comparison of the *Salmonella* phage strain KKP 3829 with five other related phage genomes exhibiting co-linearity detected by TBLASTX, using FastANI 1.3.3 software [68] and Proksee software [32]. Homologous regions detected by a TBLASTX search are connected by segments colored based on orthologous matches from query sequence fragments. The color bar shows the percentage identity of TBLASTX.

**Figure 11 ijms-24-10134-f011:**
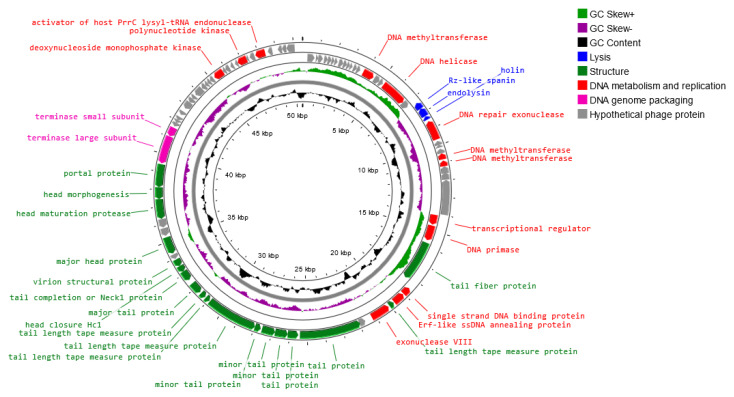
Map of the genome organization of *Salmonella* phage strain KKP 3830 generated using the Proksee software [32].

**Figure 12 ijms-24-10134-f012:**
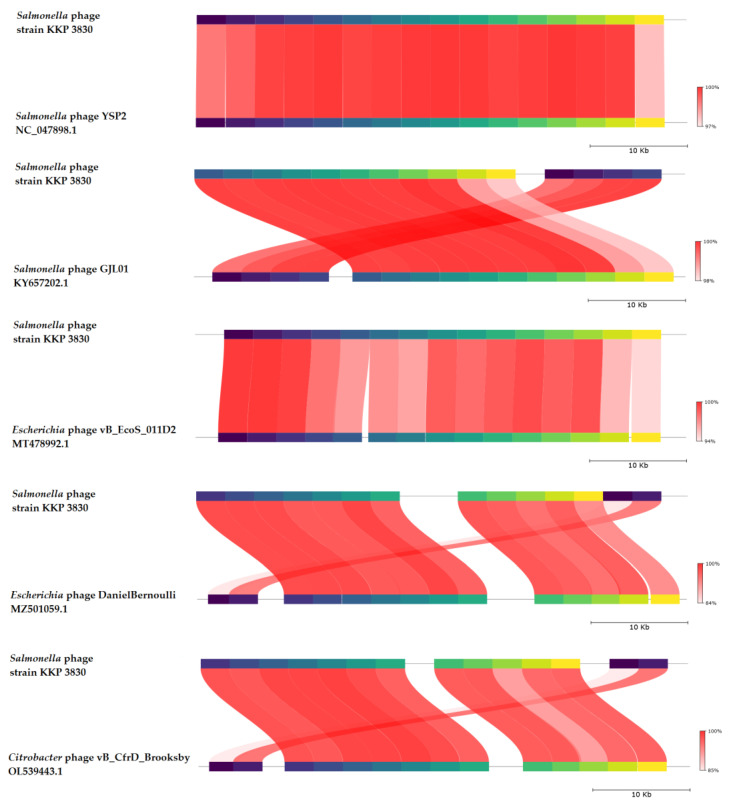
Genome sequence comparison of the *Salmonella* phage strain KKP 3830 with five other-related phage genomes exhibiting co-linearity detected by TBLASTX using FastANI 1.3.3 software [68] and Proksee software [32]. Homologous regions detected by a TBLASTX search are connected by segments colored based on orthologous matches from query sequence fragments. The color bar shows the percentage identity of TBLASTX.

**Figure 13 ijms-24-10134-f013:**
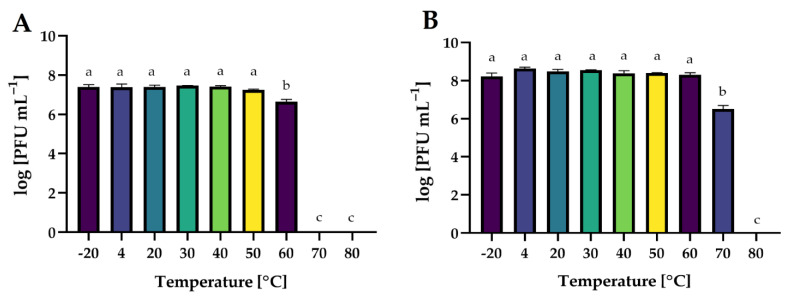
The activity of phages against bacterial host strains after exposure to a wide range of temperatures: (**A**)—*Salmonella* phage strain KKP 3829; (**B**)—*Salmonella* phage strain KKP 3830. Letters a, b, and c indicate homogenous groups at a significance level of *p* ≤ 0.05, *n* = 3.

**Figure 14 ijms-24-10134-f014:**
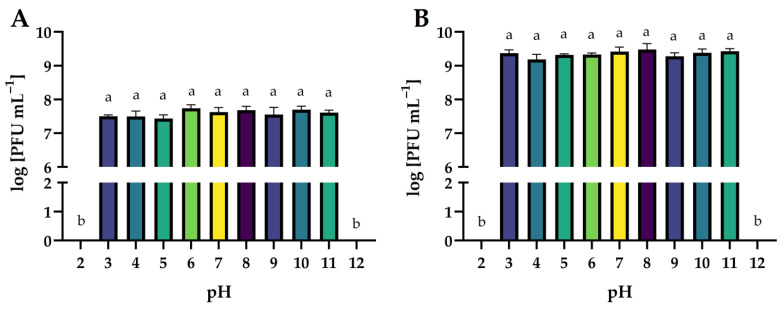
The activity of phages against bacterial host strains after exposure to a wide range of pH values: (**A**)—*Salmonella* phage strain KKP 3829; (**B**)—*Salmonella* phage strain KKP 3830. Letters a and b indicate homogenous groups at a significance level of *p* ≤ 0.05, *n* = 3.

**Figure 15 ijms-24-10134-f015:**
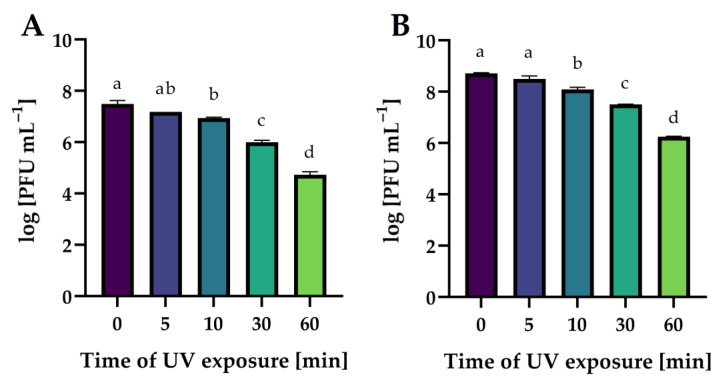
The activity of phages against bacterial host strains after different times of UV exposure: (**A**)—*Salmonella* phage strain KKP 3829; (**B**)—*Salmonella* phage strain KKP 3830. Letters a, b, c, and d indicate homogenous groups at a significance level of *p* ≤ 0.05, *n* = 3.

**Figure 16 ijms-24-10134-f016:**
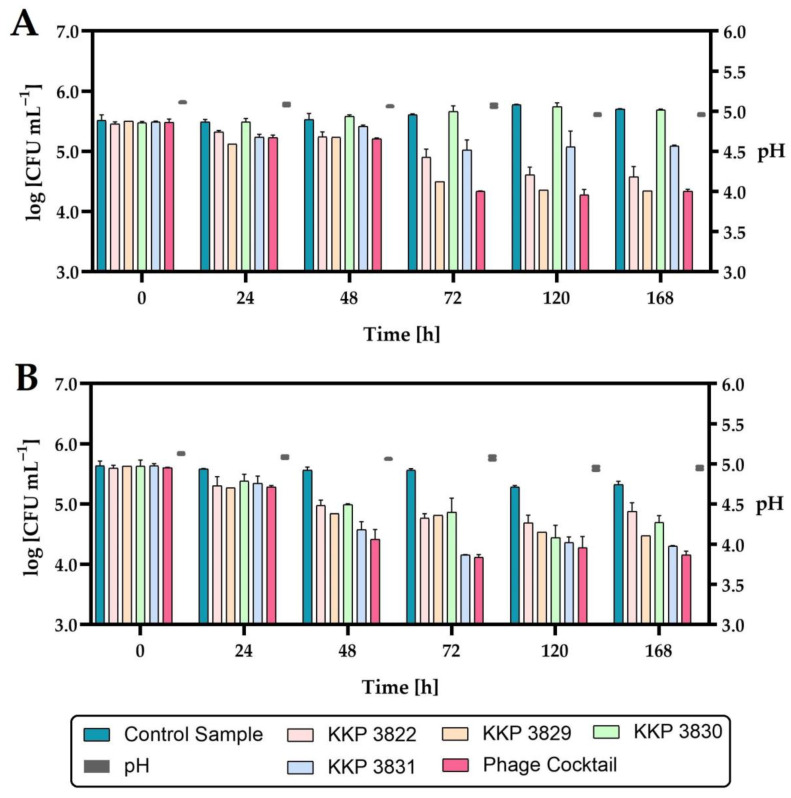
The reduction level of the *Salmonella* count in HHP-preserved carrot–mango–apple juice during storage at 4 °C after single-phage and phage-cocktail application (*n* = 3): (**A**)—*Salmonella enterica* subsp. *enterica* serovar 6,8:l,-:1,7 strain KKP 1762; (**B**)—*Salmonella* Typhimurium strain KKP 3080. The pH changes during storage are shown on the right side of the graphs.

**Figure 17 ijms-24-10134-f017:**
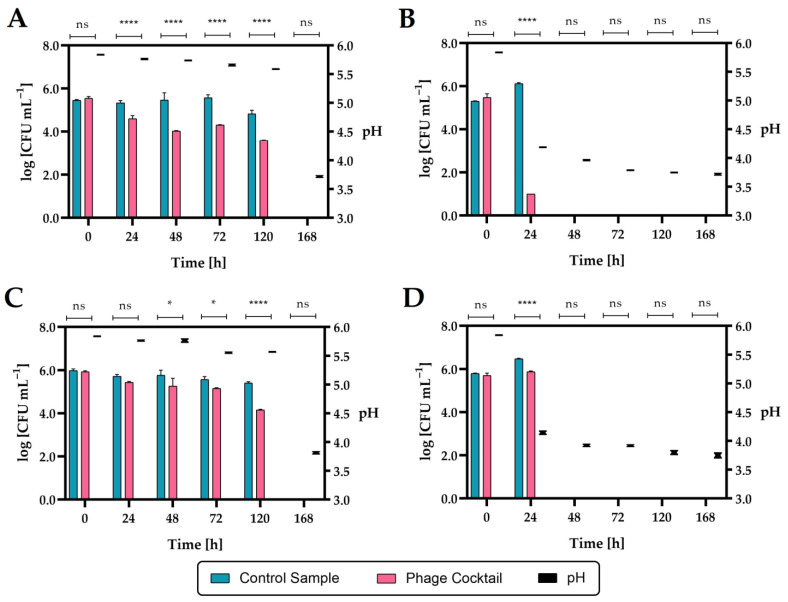
The reduction level of the *Salmonella* count in raw carrot–apple juice during storage after phage cocktail application (*n* = 3): (**A**)—*Salmonella enterica* subsp. *enterica* serovar 6,8:l,-:1,7 strain KKP 1762 at 4 °C; (**B**)—*Salmonella enterica* subsp. *enterica* serovar 6,8:l,-:1,7 strain KKP 1762 at 20 °C; (**C**)—*Salmonella* Typhimurium strain KKP 3080 at 4 °C; (**D**)—*Salmonella* Typhimurium strain KKP 3080 at 20 °C. **** means a significant difference (*p* ≤ 0.0001); * means a significant difference (*p* ≤ 0.05); ns means not significant (*p* > 0.05) of the control sample (with *Salmonella*) vs. the phage-treated samples. The pH changes during storage are shown on the right side of the graphs.

**Table 1 ijms-24-10134-t001:** *Salmonella* strains used for the isolated target bacteriophages.

Salmonella Strain Number	Full Taxonomic Name	Year of Isolation	Antibiotic Resistance Pattern
KKP 1762	Salmonella enterica subsp. enterica serovar 6,8:l,-:1,7 S. I (6,8:l,-:1,7)	2010	AMC-CPT-CT-CRO-MXF-AK-CN-TOB
KKP 3080	Salmonella enterica subsp. enterica serovar Typhimurium S. Typhimurium	2019	AMC-FEP-CTX-CPT-CT-CRO-MXF-OFX-AK-CN-TOB

Notes: AMC—amoxicillin/clavulanic acid; FEP—cefepime; CTX—cefotaxime; CPT—ceftaroline; CT—ceftolozane/tazobactam; CRO—ceftriaxone; MXF—moxifloxacin; OFX—ofloxacin; AK—amikacin; CN—gentamycin; TOB—tobramycin.

**Table 3 ijms-24-10134-t003:** Range of bacterial hosts for two newly isolated phages.

Bacterial Host Strain	GenBank Accession Number	*Salmonella* Phage Strain	
		KKP 3829	KKP 3830
*Salmonella* Strains (*n* = 54)			
*S*. Berta strain KKP 996	ON627842	+	+
*S. enterica* R strain KKP 997	MW046052	++	+
*S*. I (6,8:1,v:-) strain KKP 998	ON764274	+	–
*S*. Oranienburg strain KKP 999	ON627845	+	–
*S*. I (4,12:i:-) strain KKP 1000	ON312999	+	+
*S*. Kunduchi strain KKP 1001	MW332255	+	+
*S*. Muenster strain KKP 1002	ON340716	+	–
*S*. Hadar strain KKP 1003	ON756138	+	–
*S. enterica* R strain KKP 1004	ON627844	+	+
*S*. Senftenberg strain KKP 1005	ON627847	+	–
*S*. Derby strain KKP 1006	ON764251	–	–
*S*. Mbandaka strain KKP 1007	ON627846	+	–
*S*. I (6,8:1,v:-) strain KKP 1008	ON340717	+	–
*S*. Amsterdam strain KKP 1009	ON764277	+	+
*S*. Potsdam strain KKP 1010	ON764279	+	–
*S*. Infantis strain KKP 1039	ON764252	+	–
*S. enterica* R strain KKP 1040	ON764280	+	–
*S*. Agona strain KKP 1041	ON764253	+	+
*S*. Infantis strain KKP 1042	ON798424	+	–
*S*. Kentucky strain KKP 1043	ON764281	+	–
*S*. Muenchen strain KKP 1044	ON764287	+	–
*S*. Livingstone strain KKP 1045	ON764254	+	–
*S. enterica* R strain KKP 1113	ON775567	–	–
*S*. Mbandaka strain KKP 1169	ON764259	–	–
*S*. Abony strain KKP 1193	ON764258	+	+
*S*. Manchester strain KKP 1213	ON764805	+	–
*S*. Manchester strain KKP 1217	ON764807	+	–
*S*. Manchester strain KKP 1514	ON756136	+	–
*S*. Senftenberg strain KKP 1597	ON461374	–	–
*S. enterica* ND strain KKP 1608	ON312943	+	–
*S*. I (6,8:1,v:-) strain KKP 1610	ON313000	+	–
*S. enterica* ND strain KKP 1611	ON764857	+	–
*S*. Manchester strain KKP 1612	ON764858	+	–
*S*. Cannstatt strain KKP 1613	ON766359	+	–
*S*. Newport strain KKP 1614	ON312941	+	–
*S*. Typhimurium strain KKP 1636	ON773156	+	+
*S*. Infantis strain KKP 1761	ON798425	+	–
*S*. I (6,8:1,-:1,7) strain KKP 1762	ON340720	++ ^H^	–
*S*. Senftenberg strain KKP 1763	ON773159	+	–
*S. enterica* R strain KKP 1775	ON832663	+	+
*S*. Typhimurium strain KKP 1776	ON461376	+	+
*S.* Enteritidis strain KKP 3078	MW034593	+	+
*S.* Typhimurium strain KKP 3079	MW033548	+	+
*S.* Typhimurium strain KKP 3080	MW033536	+	++ ^H^
*S.* Typhimurium strain KKP 3081	MW033602	+	+
*S.* Enteritidis strain KKP 3814	ON732733	+	+
*S.* Enteritidis strain KKP 3815	ON732742	+	+
*S.* Enteritidis strain KKP 3816	ON756119	+	+
*S.* Enteritidis strain KKP 3817	ON756120	+	+
*S.* Enteritidis strain KKP 3818	ON756135	+	+
*S.* Typhimurium strain KKP 3819	ON732745	+	+
*S*. I (4,12:i:-) strain KKP 3820	ON732744	+	+
*S*. Abortusequi strain KKP 3821	ON732827	+	+
*S.* Sandiego strain KKP 3882	OP745459	+	+
Other Nonpathogenic *Enterobacterales* Strains (*n* = 21)			
*Citrobacter freundii* strain KKP 3655	MZ827001	–	–
*Enterobacter cloacae* strain KKP 3082	MZ827006	+	–
*Enterobacter cloacae* strain KKP 3656	OM304355	–	–
*Enterobacter cloacae* strain KKP 3684	OM281790	++	–
*Enterobacter cloacae* strain KKP 3686	OM281778	++	–
*Enterobacter ludwigii* strain KKP 3083	MZ827002	–	–
*Pantoea agglomerans* strain KKP 3651	OP978292	–	–
*Raoultella terrigena* strain KKP 3689	OK085529	–	–
*Serratia fonticola* strain KKP 3084	MZ827668	–	–
*Serratia fonticola* strain KKP 3685	OM281802	–	–
*Serratia fonticola* strain KKP 3692	OM281803	–	–
*Serratia liquefaciens* strain KKP 3654	OP978313	–	–
*Serratia marcescens* strain KKP 3687	OK103977	–	–
*Escherichia coli* strain KKP 3688	OM281784	+	–
*Escherichia coli* strain KKP 3691	OM281773	++	–
*Escherichia coli* strain KKP 3707	OM281777	+	–
*Escherichia coli* strain KKP 3800	OM250392	–	–
*Escherichia coli* strain KKP 3801	OM250391	–	–
*Escherichia coli* strain KKP 3802	OM250393	–	–
*Escherichia coli* strain KKP 3824	ON303636	–	–
*Escherichia coli* strain KKP 3825	ON303626	–	–
Other Pathogenic Gram-negative Strains (*n* = 2)			
*Pseudomonas aeruginosa* strain KKP 994	OQ302514	–	–
*Pseudomonas aeruginosa* strain KKP 1593	OK189606	–	–
Other Pathogenic Gram-positive Strains (*n* = 4)			
*Listeria monocytogenes* strain KKP 1845	OK663000	–	–
*Listeria monocytogenes* strain KKP 3270	MT990525	–	–
*Staphylococcus aureus* strain KKP 995	OQ302557	–	–
*Staphylococcus aureus* strain KKP 1082	OQ302555	–	–

Notes: For each strain of *Salmonella* serovar included in the table, an abbreviated name is used—e.g., *S.* Typhimurium means *Salmonella enterica* subsp. *enterica* serovar Typhimurium; *S*. I means *Salmonella enterica* subsp. *enterica*. “R”—rough *Salmonella* strain; “ND”—*Salmonella* serovar was not specified at the time of submitting the manuscript; “H”—original bacterial host strain; “++”—transparent plaques; “+”—turbid plaques; “–“—no plaques (non-susceptible bacterial strain). Each spot test was performed in triplicate (*n* = 3).

**Table 4 ijms-24-10134-t004:** Phage parameters calculated from the one-step growth experiments.

*Salmonella* Phage Strain	Latent Period (min)	Rise Period/Burst Time (min)	Burst Size x ± SD (PFU Cell^−1^)
KKP 3829	20	65	22 ± 0
KKP 3830	20	55	11 ± 1

**Table 5 ijms-24-10134-t005:** Phage parameters calculated from the adsorption rate experiments.

*Salmonella* Phage Strain	Adsorption at 5 min x ± SD (%)	Adsorption at 20 min x ± SD (%)	Adsorption Rate Constant (*k*) (mL min^−1^)
KKP 3829	30.0 ± 18.2	74.9 ± 1.3	2.69 × 10^9^
KKP 3830	21.1 ± 1.1	69.7 ± 1.7	1.15 × 10^9^

**Table 6 ijms-24-10134-t006:** Changes in the optical density of bacterial cultures after the addition of specific phages and values of the specific growth rate coefficient (μ).

MOI	ΔOD	μ (h^−1^)	*p*-Value (Control Culture vs. the Phage-Treated Cultures)						
			0 h	8 h	16 h	24 h	32 h	40 h	48 h
*Salmonella enterica* subsp. *enterica* serovar 6,8:l,-:1,7 strain KKP 1762									
Control Culture	0.887	0.116							
1000	N/G	N/G	ns	<0.0001 (****)	<0.0001 (****)	<0.0001 (****)	<0.0001 (****)	<0.0001 (****)	<0.0001 (****)
100	N/G	N/G	ns	<0.0001 (****)	<0.0001 (****)	<0.0001 (****)	<0.0001 (****)	<0.0001 (****)	<0.0001 (****)
10	N/G	N/G	ns	<0.0001 (****)	<0.0001 (****)	<0.0001 (****)	<0.0001 (****)	<0.0001 (****)	<0.0001 (****)
1.0	N/G	N/G	ns	<0.0001 (****)	<0.0001 (****)	<0.0001 (****)	<0.0001 (****)	<0.0001 (****)	<0.0001 (****)
0.1	0.093	0.012	ns	<0.0001 (****)	<0.0001 (****)	<0.0001 (****)	<0.0001 (****)	<0.0001 (****)	<0.0001 (****)
0.01	0.415	0.033	ns	0.0024 (**)	<0.0001 (****)	<0.0001 (****)	<0.0001 (****)	<0.0001 (****)	<0.0001 (****)
0.001	0.866	0.046	ns	<0.0001 (****)	<0.0001 (****)	<0.0001 (****)	<0.0001 (****)	<0.0001 (****)	<0.0001 (****)
0.0001	0.829	0.045	ns	<0.0001 (****)	<0.0001 (****)	<0.0001 (****)	<0.0001 (****)	<0.0001 (****)	<0.0001 (****)
*Salmonella enterica* subsp. *enterica* serovar Typhimurium strain KKP 3080									
Control Culture	0.887	0.098							
1000	0.675	0.083	ns	<0.0001 (****)	<0.0001 (****)	<0.0001 (****)	<0.0001 (****)	<0.0001 (****)	<0.0001 (****)
100	0.699	0.083	ns	<0.0001 (****)	<0.0001 (****)	<0.0001 (****)	<0.0001 (****)	<0.0001 (****)	<0.0001 (****)
10	0.706	0.083	ns	<0.0001 (****)	<0.0001 (****)	<0.0001 (****)	<0.0001 (****)	<0.0001 (****)	<0.0001 (****)
1.0	0.697	0.089	ns	<0.0001 (****)	<0.0001 (****)	<0.0001 (****)	<0.0001 (****)	<0.0001 (****)	<0.0001 (****)
0.1	0.717	0.089	ns	<0.0001 (****)	<0.0001 (****)	<0.0001 (****)	<0.0001 (****)	<0.0001 (****)	<0.0001 (****)
0.01	0.686	0.091	ns	<0.0001 (****)	<0.0001 (****)	<0.0001 (****)	<0.0001 (****)	<0.0001 (****)	<0.0001 (****)
0.001	0.676	0.091	ns	<0.0001 (****)	<0.0001 (****)	<0.0001 (****)	<0.0001 (****)	<0.0001 (****)	<0.0001 (****)
0.0001	0.764	0.097	ns	ns	0.0150 (*)	0.0001 (****)	<0.0001 (****)	<0.0001 (****)	<0.0001 (****)

N/G—no growth phase of bacterial host. **** means a significant difference (*p* ≤ 0.0001); ** means a significant difference (*p* ≤ 0.01); * means a significant difference (*p* ≤ 0.05); ns means not significant (*p* > 0.05) of the control culture (with *Salmonella*) vs. the phage-treated cultures

**Table 7 ijms-24-10134-t007:** The dimensions of the phages were determined based on the measurements of virions.

*Salmonella* Phage Strain	Total Dimension (nm)	Capsid Length (nm)	Capsid Width (nm)	Tail Length (nm)	Tail Width (nm)
KKP 3829	80.3	30.8	17.6	49.5	9.1
KKP 3830	224.9	70.4	63.6	154.5	12.3

**Table 8 ijms-24-10134-t008:** Significant differences in the number of *Salmonella* roods in juices preserved with the HHP technique and phage-treated compared to non-phage-treated juices.

Time (h)	*p*-Value (Control Sample vs. the Phage-Treated Samples)				
	*Salmonella* Phage KKP 3822	*Salmonella* Phage KKP 3829	*Salmonella* Phage KKP 3830	*Salmonella* Phage KKP 3831	Phage Cocktail
*Salmonella enterica* subsp. *enterica* serovar 6,8:l,-:1,7 strain KKP 1762					
0	ns	ns	ns	ns	ns
24	ns	0.0130 (*)	ns	ns	0.0004 (***)
48	0.0063 (**)	0.0036 (**)	ns	ns	<0.0001 (****)
72	<0.0001 (****)	<0.0001 (****)	ns	<0.0001 (****)	<0.0001 (****)
120	<0.0001 (****)	<0.0001 (****)	ns	<0.0001 (****)	<0.0001 (****)
168	<0.0001 (****)	<0.0001 (****)	ns	<0.0001 (****)	<0.0001 (****)
*Salmonella enterica* subsp. *enterica* serovar Typhimurium strain KKP 3080					
0	ns	ns	ns	ns	ns
24	0.0063 (**)	0.0197 (*)	ns	0.0042 (**)	0.0029 (**)
48	<0.0001 (****)	<0.0001 (****)	0.0002 (***)	<0.0001 (****)	<0.0001 (****)
72	<0.0001 (****)	<0.0001 (****)	<0.0001 (****)	<0.0001 (****)	<0.0001 (****)
120	<0.0001 (****)	<0.0001 (****)	<0.0001 (****)	<0.0001 (****)	<0.0001 (****)
168	0.0002 (***)	<0.0001 (****)	0.0001 (***)	<0.0001 (****)	<0.0001 (****)

**** means a significant difference (*p* ≤ 0.0001); *** means a significant difference (*p* ≤ 0.001); ** means a significant difference (*p* ≤ 0.01); * means a significant difference (*p* ≤ 0.05); ns means not significant (*p* > 0.05) of the control sample (with *Salmonella*) vs. the phage-treated samples

## Data Availability

Not applicable.

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
