# Peer review of "Newly Isolated Virulent Salmophages for Biocontrol of Multidrug-Resistant *Salmonella* in Ready-to-Eat Plant-Based Food"

_ijms, 2023, doi:10.3390/ijms241210134_

Round 1
Reviewer 1 Report
Overall, I think it is a pretty good phage research paper, and I only have several comments.
Line 21-23, 46-48 These two sentences are almost the same. You need to reword one of them.
Line 172 The word "attachment" is not suitable. Some phages can attach to their host strains but cannot form plaques due to other phage resistance mechanisms.
Line 186-198 These two phages show different host ranges. So in your discussion part, I would like to see you discuss the advantages and disadvantages of these two phages when you want to use them as biopreparation.
Line 199-220 Based on the result of one-step growth curve, we know the latent period is 20 min for both phages. Then you know you should test the adsorption rate at 20 min. Based on logic, the result of adsorption should be after the result of one-step growth curve.
Line 235-236 You use B to represent the experimental condition and burst size at the same time. It may make the reader confused.
Line 263-264 You need to clarify your controls. If you use SM buffer to contain your phages, do you add SM buffer into the culture in order to get rid of the influence of the SM buffer, or do you not add anything to your controls?
Line 320 The legend of Figure 8 is too blurry. Use vectorgraph if you can.
Line 548-541 The effect of phage on RTE food seems not good enough as biopreparation. So I would like you to discuss how to improve it in this part.
Reviewer 2 Report
This study describes the lytic activity of newly isolated Salmonella phages against MDR Salmonella in carrot-apple juice. The newly isolated phages were well characterized. However, there are few doubts that are not clearly discussed in this manuscript.
1. According to the MOI determination assay, the lytic activity of phages was highly effective and constant regardless of the MOIs. If this is true, the MOI ranges should be extended to less than 0.001 and more than 100 to see the changes. And, the OD might not be sensitive enough to see the differences, then the counting the host cells and phage titers should be better information.
2. Phage cocktail results should be displayed together with single phage treatments.
3. In Figure 16, no viable cells were observed for both control and phage cocktail after 48 h. This should be further discussed. And, clearly indicate the detection limit on the figures.
Minor comment
Table 4 – Add the time rather than Adsorption at the end.
Moderate editing of English language
Round 2
Reviewer 2 Report
Some comments have not been well responded, which are still not enough and need to be clearly stated.
1. Regardless of MOIs, no significant difference in lytic activity was observed. This is unrealistic. It needs to further interpret with their adsorption and other properties.
2. Since one of two phages was not effective, it seems to be the similar to the control though, it is not clear the synergistic effect of cocktails. In this reasons, the cocktails should be compared with the single treatment.
Moderate editing of English language
Author Response
Author's Notes to Reviewer in the attachment.

Round 3
Reviewer 2 Report
This has been well revised according to the reviewers' comments.
Moderate editing of English language required